# Mono4DGS-HDR: High Dynamic Range 4D Gaussian Splatting from Alternating-exposure Monocular Videos

**Jinfeng Liu**[1]   **Lingtong Kong**[2]   **Mi Zhou**[2]   **Jinwei Chen**[2]   **Dan Xu**[1*]

[1]The Hong Kong University of Science and Technology (HKUST)
[2]vivo Mobile Communication Co., Ltd
{jliugk,danxu}@cse.ust.hk   {ltkong,zhoumi,jinwei.chen}@vivo.com

## Abstract

We introduce Mono4DGS-HDR, the first system for reconstructing renderable 4D high dynamic range (HDR) scenes from unposed monocular low dynamic range (LDR) videos captured with alternating exposures. To tackle such a challenging problem, we present a unified framework with two-stage optimization approach based on Gaussian Splatting. The first stage learns a video HDR Gaussian representation in orthographic camera coordinate space, eliminating the need for camera poses and enabling robust initial HDR video reconstruction. The second stage transforms video Gaussians into world space and jointly refines the world Gaussians with camera poses. Furthermore, we propose a temporal luminance regularization strategy to enhance the temporal consistency of the HDR appearance. Since our task has not been studied before, we construct a new evaluation benchmark using publicly available datasets for HDR video reconstruction. Extensive experiments demonstrate that Mono4DGS-HDR significantly outperforms alternative solutions adapted from state-of-the-art methods in both rendering quality and speed. The project page for this paper is available at https://liujf1226.github.io/Mono4DGS-HDR.

## 1 Introduction

High dynamic range novel view synthesis (HDR NVS) aims to reconstruct and render HDR scenes from multi-view low dynamic range (LDR) images captured at varying exposure levels. By leveraging advanced scene representation techniques, such as Neural Radiance Field (NeRF) (Mildenhall et al., 2020) and 3D Gaussian Splatting (3DGS) (Kerbl et al., 2023), recent HDR NVS methods (Huang et al., 2022; Cai et al., 2024; Liu et al., 2025a; Wu et al., 2024a) have achieved realistic reconstruction and real-time rendering. Most of them are designed for static scenes, while HDR-Hexplane (Wu et al., 2024a) is the first to investigate on HDR NVS of dynamic scenes. However, it only validates its approach on synthetic scenes with known camera poses in a multi-camera setup. In practice, a more applicable scenario involves using a single handheld camera to capture HDR dynamic scenes in the wild. Therefore, this work focuses on recovering 4D HDR scenes from alternating-exposure monocular LDR videos with unknown camera parameters, as demonstrated in Fig. 1(a). To the best of our knowledge, no existing method has yet explored this challenging task.

Current unposed 4D reconstruction systems (Lei et al., 2025; Wang et al., 2025b; Park et al., 2025) for standard monocular videos with unchanged brightness usually leverage 2D priors from vision foundation models, including tracking, depth and optical flow, to provide scene initializations and constraints. In our input case of alternating-exposure video frames, we surprisingly observe that these 2D priors can still be effectively extracted (see in supplementary videos). However, simply extending these 4D reconstruction methods to HDR mode can lead to suboptimal results, as shown in Fig. 1(b). First, the 2D prior knowledge is still noisy and incomplete, leading to coarse and inaccurate scene initializations. Second, the varying brightness in the input video frames makes it infeasible to optimize camera poses through photometric reprojection error as Park et al. (2025). Moreover, the unreliable camera motion can further destabilize the recovery of scene dynamics and

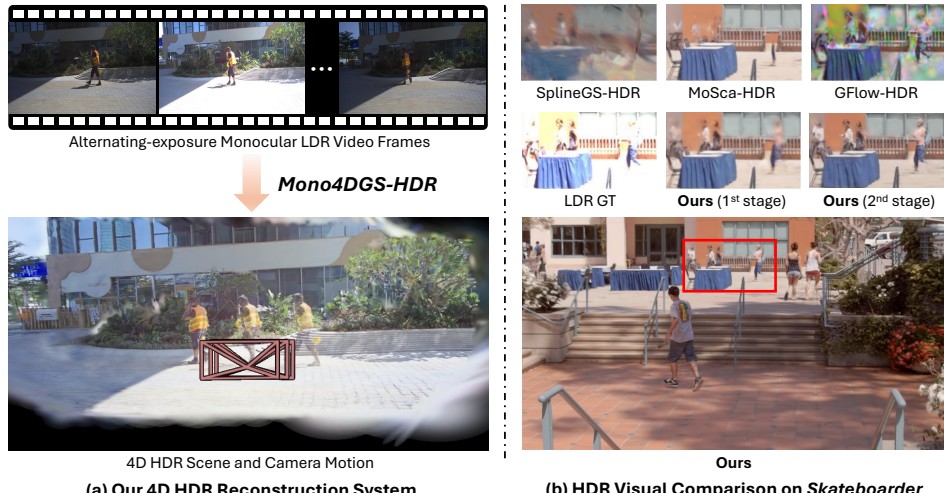

**(a) Our 4D HDR Reconstruction System**  **(b) HDR Visual Comparison on *Skateboarder***

Figure 1: (a) Our Mono4DGS-HDR can reconstruct high-quality 4D HDR scenes from unposed monocular LDR videos with alternating exposures. (b) Compared to simply extending SplineGS (Park et al., 2025), MoSca (Lei et al., 2025) and GFlow (Wang et al., 2025b) to HDR mode, our approach achieves significantly better reconstruction quality.

geometry, resulting in poor reconstruction quality. Finally, since direct HDR supervision is absent, the recovered HDR scene appearance may be temporally inconsistent and contain color artifacts.

To address the above-discussed issues, we present **Mono4DGS-HDR**, a 4D HDR reconstruction system based on Gaussian Splatting, which relies on a novel two-stage optimization approach to progressively refine the camera poses and HDR scene representation. In the first stage, inspired by SaV (Sun et al., 2024), we optimize dynamic HDR Gaussians in a 3D canonical space with an orthographic camera model, which can eliminate the need for camera poses, making HDR training video reconstruction easier and better. The learned HDR video Gaussian representation gives two advantages: (1) Consistent brightness among reconstructed HDR video frames enables reliable camera pose optimization via photometric reprojection error. (2) These video Gaussians provide a good initialization for the subsequent world Gaussian optimization. We can transform the learned video Gaussians into world space by initial camera parameters from bundle adjustment (Lei et al., 2025). The world Gaussian scaling is initialized using the invariance of the projected 2D Gaussian covariance. In the second stage, we jointly optimize camera poses and world Gaussians. The initialization from video Gaussians and the HDR photometric reprojection loss can speed up the convergence and benefit the final reconstruction quality. To further enhance the temporal consistency of HDR scene appearance, we propose a temporal luminance regularization strategy, including a flow-guided photometric loss aligning per-pixel HDR irradiance between consecutive frames, which ensures the temporal stability of HDR reconstruction and rendering.

Since our task setup has not been explored before, we build a new evaluation benchmark based on publicly available datasets (Kronander et al., 2014; Froehlich et al., 2014; Kalantari et al., 2013; Chen et al., 2021) for HDR video reconstruction, which contain real-world LDR videos with alternating exposures and synthetic HDR videos. Experiments on the datasets demonstrate that our Mono4DGS-HDR significantly outperforms alternative solutions adapted from existing 4D reconstruction systems in both rendering quality and speed. Our contributions are summarized as follows:

- We present **Mono4DGS-HDR**, the first system for reconstructing 4D HDR scenes from unposed monocular LDR videos captured with alternating exposures.

- We propose a unified framework with a two-stage optimization procedure that learns video Gaussians in the first stage, transfers the Gaussians to world space, and then optimizes world Gaussians along with camera poses in the second stage. We also conduct temporal luminance regularization to enhance HDR temporal stability.

- We construct a new benchmark for evaluation and show that our approach significantly outperforms adapted alternative solutions.

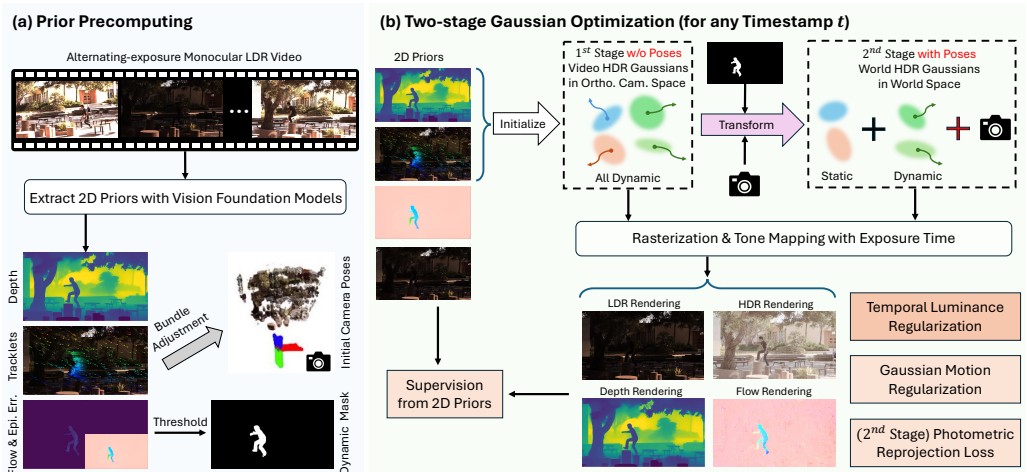

Figure 2: **Overview of Mono4DGS-HDR**. (a) We infer vision foundation models on the input alternating-exposure video to extract 2D priors, which provide scene initialization and regularization. (b) We propose a novel two-stage Gaussian optimization procedure, which includes video Gaussian training in the first stage, world Gaussian fine-tuning in the second stage, and a video-to-world Gaussian transformation strategy. The HDR Gaussians are optimized through 2D prior supervision, Gaussian motion regularization, temporal luminance regularization and HDR photometric reprojection loss.

## 2 RELATED WORK

**Dynamic Reconstruction and View Synthesis.** Dynamic scene reconstruction and view synthesis has experienced a great breakthrough by the advent of NeRF and 3DGS. Early dynamic NeRF methods usually represent the 4D scenes by time-varying NeRFs (Park et al., 2023; Gao et al., 2021; Li et al., 2022), or a canonical space NeRF with deformation field (Guo et al., 2023; Park et al., 2021a;b; Fang et al., 2022; Xian et al., 2021; Pumarola et al., 2021). Recent methods begin to extend the efficient 3DGS representation to dynamic scenes. They model dynamic contents by learning Gaussian deformation (Yang et al., 2024b; Wu et al., 2024b; Bae et al., 2024), motion trajectories (Li et al., 2024a; Lin et al., 2024; Lee et al., 2024; Yoon et al., 2025), or direct 4D Gaussian primitives (Yang et al., 2024a; Duan et al., 2024). While most of these works take as input synchronized multi-view videos to make problem easier, a growing body of works (Liu et al., 2023; Sun et al., 2024; Wang et al., 2025a; Lei et al., 2025; Wang et al., 2025b; Park et al., 2025; Stearns et al., 2024; Liu et al., 2025b) address the more practical and challenging scenario of monocular videos. Among them, RoDynRF (Liu et al., 2023), MoSca (Lei et al., 2025), SplineGS (Park et al., 2025) and GFlow (Wang et al., 2025b) are notable for handling unposed monocular videos, utilizing the 2D prior knowledge of tracking, depth and optical flow. However, they are designed for standard videos with consistent brightness and struggle with the challenges posed by alternating exposures. In this work, we introduce a novel framework to explicitly solve on such varying-exposure videos.

**HDR Novel View Synthesis.** Existing HDR NVS approaches generally fall into two categories. The first leverages noisy RAW sensor data (Mildenhall et al., 2022; Wang et al., 2024; Jin et al., 2024; Singh et al., 2024; Li et al., 2024b), which is particularly suited for low-light or nighttime scenarios. This work is more related to the second category, which utilizes multi-exposure and multi-view LDR images for training (Huang et al., 2022; Jun-Seong et al., 2022; Wu et al., 2024a; Cai et al., 2024; Wu et al., 2024c; Liu et al., 2025a). HDR-NeRF (Huang et al., 2022) pioneers the idea of learning HDR radiance fields by regressing HDR irradiance rather than LDR color and introduces tone-mapping MLPs to model the camera response function (CRF). Later works, including HDR-Plenoxels (Jun-Seong et al., 2022) and HDR-GS (Cai et al., 2024), improve the efficiency by adopting alternative scene representations such as Plenoxels (Fridovich-Keil et al., 2022) and 3DGS. GaussHDR (Liu et al., 2025a) learns unified 3D and 2D local tone mapping for stable and high-quality HDR reconstruction. Casual3DHDR (Gong et al., 2025) applies continuous-time trajectory to jointly optimize camera poses, CRF and exposure times from casually captured videos. However, most of these

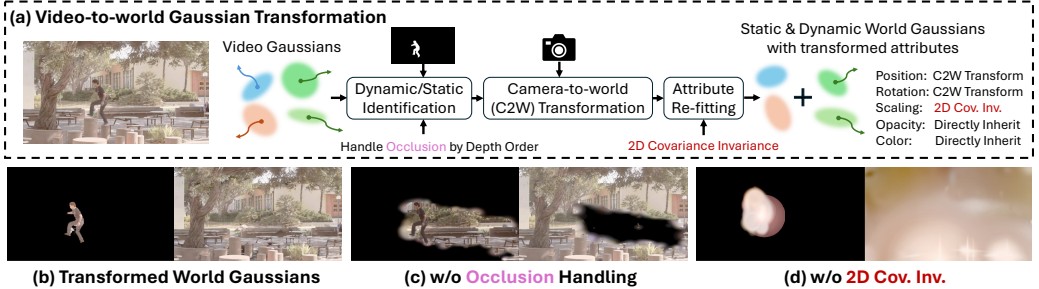

Figure 3: (a) Our video-world Gaussian Transformation Strategy, including dynamic/static identification, attribute transformation and re-fitting. (b) Example of transformed dynamic/static world Gaussians. (c) Without occlusion handling, the dynamic/static separation is inaccurate. (d) Without 2D covariance invariance (directly inherit scaling), the world Gaussians have unreasonable scales.

methods are designed for static scenes. Although HDR-HexPlane (Wu et al., 2024a) extends this task to dynamic scenes with HexPlane (Cao & Johnson, 2023) representation, it requires known camera poses and is validated on a multi-camera setting. Unlike it, our work is the first to tackle the more practical setting of unposed monocular videos.

**HDR Video Reconstruction.** HDR videos can be directly collected using dedicated hardware like scanline exposure/ISO (Choi et al., 2017; Heide et al., 2014) and beam splitters (Tocci et al., 2011; McGuire et al., 2007). But they are often impractical due to their sophisticated designs and high cost. Hence, reconstructing HDR videos from alternative-exposure LDR videos is investigated, which aligns neighboring frames to a reference frame and then merging the aligned images to an HDR image (Kang et al., 2003; Kalantari et al., 2013). Later deep learning based works (Kalantari & Ramamoorthi, 2019; Chen et al., 2021; Chung & Cho, 2023; Xu et al., 2024) mainly use a flow network for alignment and a weight network for merging. However, the HDR data for supervised training is rare, limiting their generalization ability. Moreover, these methods can only recover training HDR videos. They can neither synthesize LDR images at new exposure levels nor support NVS. In contrast, our method reconstructs the whole 4D HDR scene in a self-supervised manner, enabling novel-view rendering of both HDR videos and LDR videos with controllable exposures.

## 3 METHODOLOGY

In our monocular 4D HDR reconstruction setup, the input is an unposed monocular video consisting of alternating-exposure LDR frames $\{L_t\}_{t=1}^{N_f}$, where $N_f$ is the frame number. Consider 2-exposure case, $\{L_1, L_3, ...\}$ are captured with a short exposure time $\Delta t_s$ and $\{L_2, L_4, ...\}$ with long exposure time $\Delta t_l$. Other exposure patterns (e.g., 3-exposure) can be similarly extended. Our goal is to recover the renderable 4D HDR scene along with unknown camera parameters through Gaussian splatting. The overview of our Mono4DGS-HDR is illustrated in Fig. 2. In the following, we first briefly introduce the preliminaries of HDR GS and Gaussian dynamics in Sec. 3.1. Then, we detail each step and component of our system in Sec. 3.2 and Sec. 3.3.

### 3.1 PRELIMINARIES

**HDR Gaussian Splatting.** HDR Gaussian Splatting (Cai et al., 2024; Liu et al., 2025a) represents an HDR scene using a set of anisotropic 3D HDR Gaussians, which is the same as 3DGS (Kerbl et al., 2023) except replacing LDR color $c \in [0, 1]$ with HDR irradiance $e \in [0, +\infty)$. Besides, there is a logarithmic-domain tone mapper $\phi$ to map the HDR irradiance to LDR color at a given exposure time $\Delta t$, denoted as $c = \phi(\log(e\Delta t))$. Details about 3DGS are provided in Sec. A.1.1.

**Gaussian Dynamics.** Unlike those works (Wu et al., 2024b; Yang et al., 2024b; Liu et al., 2025b) that leverage deformation MLPs to model temporal changes and thus slow down rendering speed, we choose to explicitly parameterize the motion of dynamic Gaussians with trajectory functions following Park et al. (2025); Li et al. (2024a), which preserves the rendering speed of 3DGS and enables seamless transformation from video Gaussians to world Gaussians. Specifically, we use the cubic Hermite spline function to represent the position trajectory of each Gaussian (see in Sec. A.1.2),

containing $N_c$ control points. For Gaussian rotation, we employ a cubic polynomial function to represent quaternion as $r(t) = \sum_{j=0}^{3} a_j t^j$, where $\{a_j | a_j \in \mathbb{R}^4\}_{j=0}^{3}$ are polynomial coefficients. The scaling, opacity, and color of each Gaussian are assumed to be time-invariant for simplicity.

## 3.2 PRIOR PRECOMPUTE

Since monocular 4D reconstruction is highly ill-posed, we leverage the prior knowledge inferred by vision foundation models as Park et al. (2025); Lei et al. (2025) to provide scene initialization and regularization. As shown in Fig. 2(a), we use off-the-shelf foundation models to obtain: (1) Video depth estimations (Hu et al., 2025); (2) Sparse long-term 2D pixel trajectories (Xiao et al., 2024); (3) Per-frame epipolar error maps (Liu et al., 2023) computed from dense optical flow predictions (Teed & Deng, 2020), that can identify the dynamic foreground masks by thresholding. Note that in our input of alternating-exposure LDR frames, we should compute the optical flow between two frames at the same exposure level. For video depth estimation and long-term tracklet prediction, we just feed the whole frame sequence to the models. With these priors, we can also conduct bundle adjustment using static tracklets to obtain initial camera parameters as in MoSca (Lei et al., 2025).

## 3.3 TWO-STAGE GAUSSIAN OPTIMIZATION

Although the 2D priors can be effectively extracted from our input alternating-exposure videos, they are still noisy and insufficient, which results in coarse scene initializations. Therefore, we propose a unified framework with a novel two-stage Gaussian optimization procedure, which consists of video Gaussian training in the first stage, world Gaussian fine-tuning in the second stage, and a transformation strategy from video Gaussians to world Gaussians.

### 3.3.1 VIDEO HDR GAUSSIANS

Inspired by SaV (Sun et al., 2024), we learn a set of fully dynamic video HDR Gaussians in orthographic camera coordinate space at the first stage. For a 3D point $p^v = [x^v, y^v, z^v]$ in this space, $(x^v, y^v) \in [-1, 1]^2$ is actually the normalized coordinate of its projected pixel position, while $z^v$ is the depth. In this representation, we should use an orthographic camera model for rasterization and replace the original projection Jacobian $J$ in 3DGS with $J_{\text{ortho}}$. We can lift the track/depth priors to initialize video Gaussians. Please refer to Sec. A.1.3 for more details. In a word, video Gaussian representation enables us to treat camera motion and object motion uniformly as the motion of dynamic Gaussians and eliminate the need of camera parameters. Consequently, we can fit LDR training frames and recover HDR training video more efficiently. The resulting HDR video Gaussians provide a reliable foundation for subsequent world Gaussian and camera pose refining.

### 3.3.2 VIDEO-TO-WORLD GAUSSIAN TRANSFORMATION

The learned video Gaussians are in pseduo-3D space, which cannot represent the actual 3D geometry in the world. Thus, we design a video-to-world Gaussian transformation strategy (see in Fig. 3(a)), to make the initialization from video Gaussians reasonable for further world Gaussian optimization.

**Dynamic & Static Identification with Occlusion Handling.** First, we need to identify whether a dynamic video Gaussian is static or dynamic in world space. To this end, we leverage dynamic masks $\mathcal{M} = \{M_t\}_{t=1}^{N_f}$ derived from epipolar error maps, where 1 and 0 in $M_t$ indicate dynamic and static regions, respectively. Concretely, we project each video Gaussian trajectory $\mathcal{G}^v = \{\mu_t^v | \mu_t^v = [x_t^v, y_t^v, z_t^v]\}_{t=1}^{N_f}$ to image plane and count the occurrences $N_d$ when it falls into dynamic regions:

$$N_d = \sum\nolimits_{t=1}^{N_f} \mathcal{I}[M_t(x_t^v, y_t^v) \cdot (1 - o_t) = 1], \quad o_t = \mathcal{I}[z_t^v > \widetilde{D}_t(x_t^v, y_t^v)], \tag{1}$$

where $\mathcal{I}[\cdot]$ is the indicator function, and $o_t = 1$ means the Gaussian is occluded at time $t$ (i.e., its depth is larger than the rendered depth $\widetilde{D}_t$). If $N_d/N_f$ is larger than a pre-defined threshold (e.g., 0.1), we consider this video Gaussian as dynamic in the world space and static otherwise.

**Gaussian Position & Rotation Transform.** Then, we transform the video Gaussian, with per-frame positions $\{\mu_t^v\}_{t=1}^{N_f}$ and rotations $\{R_t^v\}_{t=1}^{N_f}$, into world space using the initial camera intrinsics $\hat{K}$ and extrinsics $\{[\hat{R}_t|\hat{T}_t]\}_{t=1}^{N_f}$ obtained from bundle adjustment. Let $\pi_{\hat{K}}(\cdot)$ be the projection function from

camera space to image space with intrinsics $\hat{K}$ and $\pi_{\hat{K}}^{-1}(\cdot)$ be the inverse. We can transform as $\mu_t^w = \hat{R}_t \pi_{\hat{K}}^{-1}(\mu_t^v) + \hat{T}_t$ and $R_t^w = \hat{R}_t R_t^v$, where $\mu_t^w$ and $R_t^w$ are the Gaussian position and rotation at time $t$ in world space, respectively. For static Gaussian, we simply set the time-independent world position as $\mu^w = \frac{1}{N_f} \sum_{t=1}^{N_f} \mu_t^w$ and rotation as $R^w = \mathrm{Avg}_R(\{R_t^w\}_{t=1}^{N_f})$, where $\mathrm{Avg}_R(\cdot)$ is the rotation quaternion averaging method proposed by Markley et al. (2007). For dynamic Gaussian, we re-sample control points from the world position trajectory $\{\mu_t^w\}_{t=1}^{N_f}$ for spline function initialization and re-fit the polynomial coefficients for rotation quaternion trajectory using least squares.

**Gaussian Opacity & Color Inheriting.** The attributes of opacity and HDR color are directly inherited from the video Gaussians, no matter static or dynamic, since they are intuitively space-invariant.

**Gaussian Scaling Re-fitting by 2D Covariance Invariance.** Note that the scale difference between camera coordinate space and world space should be carefully handled. To obtain initial world Gaussians with rational scales, we propose to re-fit Gaussian scaling based on the invariance of 2D covariance, which is motivated by the fact that the projected 2D Gaussians should have consistent shapes and sizes before and after transformation. The projected 2D covariances of video and world Gaussians at time $t$, denoted as $\Sigma_t'^v$ and $\Sigma_t'^w$, can be derived as:

$$\Sigma_t'^v = [J_{\mathrm{ortho}} W_t^v R_t^v S^v (J_{\mathrm{ortho}} W_t^v R_t^v S^v)^\top]_{2\times 2}, \quad \Sigma_t'^w = [J W_t^w R_t^w S^w (J W_t^w R_t^w S^w)^\top]_{2\times 2}, \quad (2)$$

where $S^v$ and $S^w$ are the time-invariant scaling matrices of video and world Gaussians, and $[\cdot]_{2\times 2}$ means skipping the third row and column. For the viewing transformations, we have $W_t^v = E$ (identity matrix) and $W_t^w = \hat{R}_t^{-1}$. Now, we can obtain the initial world Gaussian scaling by solving the optimization problem $\min_{S^w} \sum_{t=1}^{N_f} ||\Sigma_t'^v - \Sigma_t'^w||_2$. We apply it to both static and dynamic Gaussians, and solve it using gradient descent, that can converge within 1000 iterations in 1 minute.

After above video-to-world Gaussian transformation, we can step into the second stage to refine the world Gaussians together with camera parameters. Due to the appropriate initialization from video Gaussians, the optimization in world space can converge fast and stably.

### 3.3.3 OPTIMIZATION STRATEGY

We render from the fully dynamic video Gaussians in the first stage, and from the union of static and dynamic world Gaussians in the second stage. At each training iteration, we randomly sample a query timestamp $t \in \{1, 2, ..., N_f\}$. The rendered variables at $t$ include the HDR image $\widetilde{H}_t$, the depth map $\widetilde{D}_t$, and the flow/track map $\widetilde{F}_{t \to t'}$, where $t'$ is the destination timestamp. To obtain $\widetilde{F}_{t \to t'}$, we can rasterize the 3D movements of all Gaussians in camera space from $t$ and $t'$ and project to the image plane. The HDR rendering can be then converted to LDR image $\widetilde{L}_t$ with tone-mapping MLPs and exposure time. The following descriptions apply to both stages unless otherwise specified.

**Supervision from 2D Priors.** We supervise the scene geometry, appearance and dynamics with input LDR frames and precomputed 2D priors, including LDR RGB loss $\mathcal{L}_{\mathrm{rgb}}$, depth loss $\mathcal{L}_{\mathrm{dep}}$, flow/track loss $\mathcal{L}_{\mathrm{track}}$ and unit exposure loss $\mathcal{L}_{\mathrm{ue}}$. Please refer to Sec. A.1.4 for details.

**Gaussian Motion Regularization.** We also regularize the dynamic Gaussians to ensure smooth and plausible motions as Lei et al.

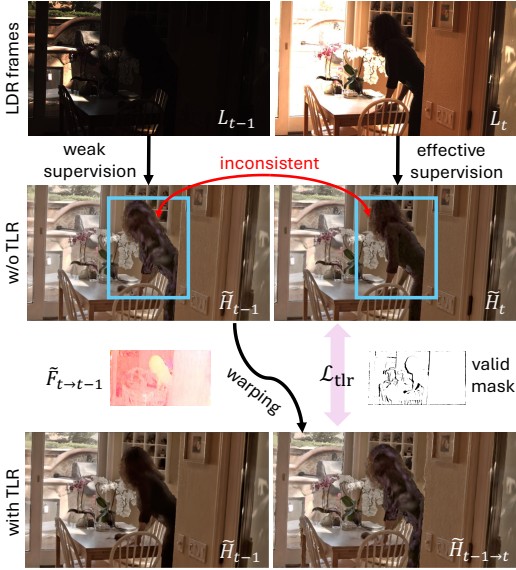

Figure 4: Temporal luminance regularization for temporally consistent HDR appearance.

(2025); Sun et al. (2024); Wang et al. (2025a), including as-rigid-as-possible loss $\mathcal{L}_{\mathrm{arap}}$, velocity regularization $\mathcal{L}_{\mathrm{vel}}$, and acceleration regularization $\mathcal{L}_{\mathrm{acc}}$ (see in Sec. A.1.5 for details).

Table 1: Quantitative comparisons on the test frames of Syn-Exp-3 scenes. Metrics are averaged over all scenes. LDR-OE and LDR-NE denote the LDR results with observed and novel exposures, respectively. HDR denotes the HDR results. FPS is measured at $864 \times 480$ resolution. † We use our initial camera parameters from bundle adjustment as the required camera inputs for GaussHDR (Liu et al., 2025a) and HDR-HexPlane (Wu et al., 2024a). ‡ We extend SplineGS (Park et al., 2025) and MoSca (Lei et al., 2025) to HDR mode for fair comparison.

| Method | LDR-OE | | | LDR-NE | | | HDR | | | | Training time | FPS |
|---|---|---|---|---|---|---|---|---|---|---|---|---|
| | PSNR↑ | SSIM↑ | LPIPS↓ | PSNR↑ | SSIM↑ | LPIPS↓ | PSNR↑ | SSIM↑ | LPIPS↓ | TAE↓ | | |
| GaussHDR† | 29.51 | 0.858 | 0.167 | 28.96 | 0.863 | 0.165 | 31.25 | 0.891 | 0.105 | 0.089 | **1h** | 51 |
| HDR-HexPlane† | 29.26 | 0.806 | 0.186 | 28.72 | 0.812 | 0.189 | 29.60 | 0.839 | 0.120 | 0.155 | **1h** | 1 |
| SplineGS-HDR‡ | 17.59 | 0.661 | 0.495 | 16.60 | 0.646 | 0.517 | 17.82 | 0.677 | 0.447 | 1.188 | 1.5h | 79 |
| MoSca-HDR‡ | 34.08 | 0.898 | 0.098 | 33.92 | 0.910 | 0.092 | 36.89 | 0.952 | 0.053 | 0.059 | 1.5h | 82 |
| **Mono4DGS-HDR (Ours)** | **34.75** | **0.904** | **0.086** | **34.54** | **0.915** | **0.081** | **37.64** | **0.959** | **0.042** | **0.057** | 1.5h | **161** |

**Temporal Luminance Regularization (TLR).** The appearance supervision is only from LDR frames, which may cause unstable HDR luminance across time, especially for dynamic scenes, since dynamic Gaussians tend to float around the surfaces of moving objects. The supervision is effective at the times when the corresponding dynamic contents are properly exposed, but it is weak when over/under-exposed. The former case leads to correctly positioned dynamic Gaussians while the latter can result in floaters above dynamic object surfaces. Consequently, the HDR appearance of dynamic contents may vary significantly at different times, as shown in Fig. 4. To address this issue, we propose temporal luminance regularization using flow-guided photometric loss to align per-pixel HDR luminance between consecutive frames. Given two adjacent timestamps $t - 1$ and $t$, we can warp the HDR rendering $\widetilde{H}_{t-1}$ to time $t$ using optical flow, generating $\widetilde{H}_{t-1\to t}$. Then, we can compute the loss as:

$$\mathcal{L}_{\text{tlr}} = \left| V_{t\to t-1} \odot \frac{\widetilde{H}_{t-1\to t} - \widetilde{H}_t}{\widetilde{H}_{t-1\to t} + \widetilde{H}_t} \right|_1, \qquad (3)$$

where $V_{t\to t-1}$ is a valid mask derived from depth order to exclude occluded pixels. By normalizing with $\widetilde{H}_{t-1\to t} + \widetilde{H}_t$, we can eliminate the influence of HDR irradiance scale. At query time $t$, we choose both $t - 1$ and $t + 1$ as the destinations to obtain $\mathcal{L}_{\text{tlr}}$. Since we cannot access optical flow priors between adjacent frames at different exposures, we use the rendered flow maps $\widetilde{F}_{t\to t-1}$ and $\widetilde{F}_{t\to t+1}$(stop gradients) for warping. To this end, we only apply TLR after the ending of Gaussian densification to ensure reliable flow rendering. With TLR, the learned dynamic contents at well-supervised times can propagate to the poorly-supervised times, leading to temporally consistent HDR appearance, as shown in Fig. 4.

**HDR Photometric Reprojection Loss.** Previous works like SplineGS (Park et al., 2025) employ a photometric reprojection loss $\mathcal{L}_{\text{pr}}$ (Godard et al., 2019; Liu et al., 2024b;a) in monocular videos to optimize depth and camera poses together, which is not suitable for our alternating-exposure inputs. However, we can leverage the recovered HDR training video from the first stage for this purpose. With $\mathcal{L}_{\text{pr}}$ in Sec. A.1.6, we jointly refine camera poses and world Gaussians in the second stage.

**Overall Loss.** The overall objective is $\mathcal{L} = \lambda_{\text{rgb}}\mathcal{L}_{\text{rgb}} + \lambda_{\text{ue}}\mathcal{L}_{\text{ue}} + \lambda_{\text{dep}}\mathcal{L}_{\text{dep}} + \lambda_{\text{track}}\mathcal{L}_{\text{track}} + \lambda_{\text{arap}}\mathcal{L}_{\text{arap}} + \lambda_{\text{vel}}\mathcal{L}_{\text{vel}} + \lambda_{\text{acc}}\mathcal{L}_{\text{acc}} + \lambda_{\text{tlr}}\mathcal{L}_{\text{tlr}} + \lambda_{\text{pr}}\mathcal{L}_{\text{pr}}$, where $\lambda$'s are the corresponding weights.

**Gaussian Densification.** We follow the same densification strategy as 3DGS for all Gaussians at both stages. In the second stage, we prune dynamic world Gaussians every certain number of iterations to remove those densified mistakenly. We can project dynamic Gaussian trajectories to camera coordinate space and apply the dynamic/static identification method in Sec. 3.3.2 for filtering.

## 4 EXPERIMENTS

### 4.1 EXPERIMENTAL SETTINGS

**Datasets and Evaluation Metrics.** Since our task has never been explored before, we create a new evaluation benchmark base on publicly available datasets (Kronander et al., 2014; Froehlich et al., 2014; Kalantari et al., 2013; Chen et al., 2021) for HDR video reconstruction, resulting in 25 dynamic scenes in total. Each scene contains an alternating-exposure video clip with 50-100 frames. We classify them into three categories: (1) Syn-Exp-3: 9 synthetic scenes with 3 exposure levels and

Table 2: Quantitative comparisons on the train frames of Real-Exp-2 scenes and the test frames of Real-Exp-3 scenes. Metrics are averaged over all scenes. OE denotes the observed-exposure results. † We use our initial camera parameters from bundle adjustment as camera inputs for GaussHDR (Liu et al., 2025a) and HDR-HexPlane (Wu et al., 2024a). ‡ We extend GFlow (Wang et al., 2025b), SplineGS (Park et al., 2025) and MoSca (Lei et al., 2025) to HDR mode for fair comparison.

| Method | Real-Exp-2 (train frames) | | | | Real-Exp-3 (test frames) | | | |
|---|---|---|---|---|---|---|---|---|
| | LDR-OE | | | HDR-TAE↓ | LDR-OE | | | HDR-TAE↓ |
| | PSNR↑ | SSIM↑ | LPIPS↓ | | PSNR↑ | SSIM↑ | LPIPS↓ | |
| GaussHDR† | 24.77 | 0.895 | 0.106 | 0.073 | 24.98 | 0.858 | 0.117 | 0.096 |
| HDR-HexPlane† | 29.30 | 0.900 | 0.106 | 0.154 | 18.49 | 0.627 | 0.241 | 0.705 |
| GFlow-HDR‡ | 27.71 | 0.908 | 0.094 | 0.426 | - | - | - | - |
| SplineGS-HDR‡ | 21.46 | 0.721 | 0.334 | 1.282 | 13.64 | 0.506 | 0.505 | 0.707 |
| MoSca-HDR‡ | 30.28 | 0.915 | 0.074 | 0.054 | 27.23 | 0.872 | 0.084 | 0.076 |
| **Mono4DGS-HDR (Ours)** | **31.82** | **0.928** | **0.052** | **0.046** | **27.65** | **0.876** | **0.081** | **0.067** |

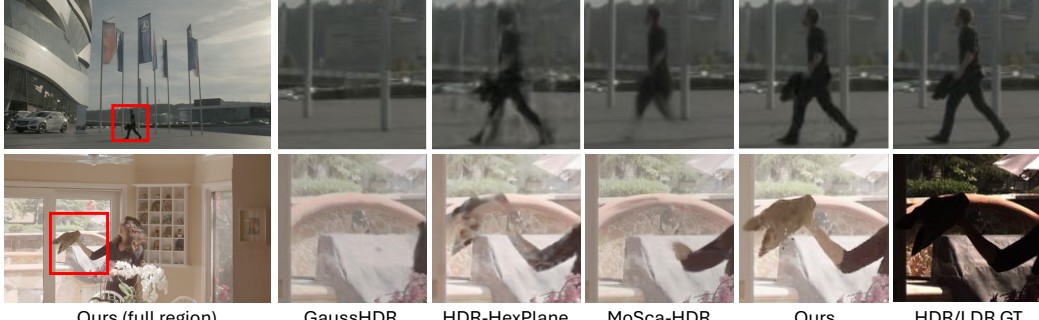

Figure 5: HDR visual comparisons on train/test frames. Our method achieves superior quality.

HDR GTs; (2) Real-Exp-3: 8 real-world scenes with 3 exposure levels; and (3) Real-Exp-2: 8 real-world scenes with 2 exposure levels. Details about the datasets are given in Sec. A.2.1. For quantitative evaluation, we utilize PSNR (higher is better), SSIM (higher is better), and LPIPS (Zhang et al., 2018) (lower is better) metrics. Following HDR-NeRF (Huang et al., 2022), we quantitatively evaluate HDR results in the $\mu$-law (Kalantari & Ramamoorthi, 2017) domain and qualitatively show HDR results via Photomatix pro. We additionally introduce HDR-TAE (lower is better) metric to measure the temporal stability of rendered HDR videos, which is detailed in Sec. A.2.2.

**Implementation Details.** For prior precomputing, we utilize DepthCrafter (Hu et al., 2025) for video depth estimation, SpatialTracker (Xiao et al., 2024) for track prediction and RAFT (Teed & Deng, 2020) for optical flow estimation. Models are trained for 4K iterations in the first stage and 11K iterations in the second stage. For the camera poses of test frames, we interpolate from the neighboring train frames. Please refer to Sec. A.2.3 for more details.

## 4.2 PERFORMANCE COMPARISON

**Baselines.** We compare our method with several baselines, which can be categorized into two groups: (1) *HDR NVS methods*, including GaussHDR (Liu et al., 2025a) for static scenes and HDR-HexPlane (Wu et al., 2024a) for dynamic scenes but in multi-camera setting; (2) *standard unposed 4D monocular reconstruction methods*, including MoSca (Lei et al., 2025), SplineGS (Park et al., 2025) and GFlow (Wang et al., 2025b). For fair comparison, we extend the second group to HDR mode by employing HDR color and applying the same tone-mapper MLPs as ours. Note that GFlow cannot support for time interpolation of Gaussians, so we can only evaluate it on the training frames. For the first group, since they require known camera poses, we use our initial camera parameters as their camera inputs. We train all baseline methods for the same 15K iterations as ours.

**Quantitative & Qualitative Comparison.** The quantitative results of synthetic and real scenes are listed in Table 1 and Table 2, respectively. More comparisons with GFlow on training frames are provided in Appendix (Table 5). We can see that our method outperforms all baseline methods by a large margin across all tracks. GaussHDR and HDR-HexPlane perform poorly owing to their incapability of handling dynamic scenes and monocular videos, respectively. GFlow fails to recover HDR scenes since it optimizes per-frame Gaussians independently to overfit the LDR observations.

Table 3: Quantitative ablation results on the test frames of Real-Exp-3 and Syn-Exp-3 scenes. V2W denotes the video-to-world Gaussian transformation. All experiments are trained for 15K iteration. All the metrics listed here represent PSNR except HDR-TAE.

|  | Method | Real-Exp-3 | | Syn-Exp-3 | | | |
|---|---|---|---|---|---|---|---|
|  |  | LDR-OE | HDR-TAE | LDR-OE | LDR-NE | HDR | HDR-TAE |
| (a) | w/o Video Gaussian Initialization | 26.47 | 0.068 | 33.62 | 33.38 | 36.07 | **0.057** |
| (b) | w/o Occlusion Handling in V2W | 27.29 | 0.069 | 34.40 | 34.17 | 37.22 | 0.059 |
| (c) | w/o 2D Covariance Invariance in V2W | 27.34 | 0.068 | 34.31 | 34.21 | 37.25 | **0.057** |
| (d) | w/o HDR Photometric Reprojection Loss | 27.26 | 0.070 | 34.38 | 34.17 | 37.33 | 0.059 |
| (e) | w/o Temporal Luminance Regularization | 27.63 | 0.082 | 34.71 | 34.49 | 37.58 | 0.071 |
| (f) | Mono4DGS-HDR (Full model) | **27.65** | **0.067** | **34.75** | **34.54** | **37.64** | 0.057 |

SplineGS behaves the worst since it heavily relies on the photometric reprojection loss to recover camera motion, which is infeasible in the presence of alternating-exposure LDR frames. MoSca achieves the second best performance due to its global Gaussian fusion ability, but still inferior to our method. Overall, our Mono4DGS-HDR demonstrates state-of-the-art performance in terms of temporal stability, rendering quality and speed. For qualitative results, we provide HDR visual comparisons of train/test frames in Fig. 5. More examples are exhibited in Appendix (Fig. 8). We also present visual results under fix-view-change-time and fix-time-change-view settings in Appendix (Fig. 9 and Fig. 10). It can be observed that our method generates higher-quality HDR renderings with finer details and fewer artifacts than baselines.

## 4.3 ABLATION STUDY

In this part, we conduct ablation studies to validate the effectiveness of our key designs.

**Effect of Video Gaussian Initialization.** To evaluate the effectiveness of the initialization from video Gaussians, we remove the video Gaussian stage and directly train Gaussians in the world space, where we initialize the world Gaussians with prior track and depth information. As listed in Table 3(a), the PSNR performance drops significantly with more than 1dB. Fig. 6(a) also shows noticeable visual degradation. These demonstrate the importance of our video Gaussian initialization. Compared to track/depth lifting, the video Gaussians provide not only accurate position priors but also meaningful rotation, scaling, opacity and color priors, which greatly facilitate the subsequent world Gaussian optimization.

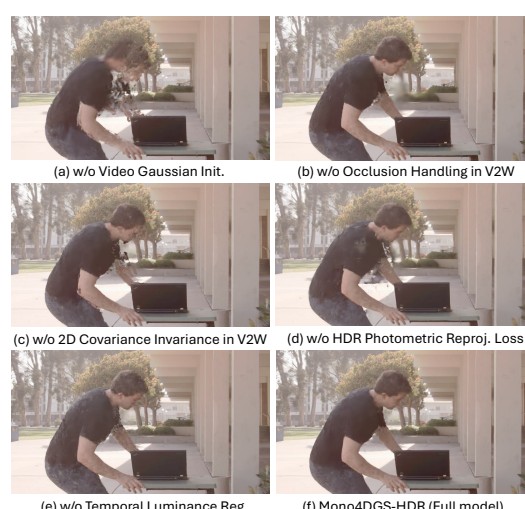

(a) w/o Video Gaussian Init.  (b) w/o Occlusion Handling in V2W

(c) w/o 2D Covariance Invariance in V2W  (d) w/o HDR Photometric Reproj. Loss

(e) w/o Temporal Luminance Reg.  (f) Mono4DGS-HDR (Full model)

Figure 6: Qualitative ablation results.

**Effect of Occlusion Handling & 2D Covariance Invariance in V2W.** We further explore the role of occlusion handling and 2D covariance invariance in our video-to-world Gaussian transformation by ablating them separately. As shown in Fig. 3(b)(c)(d), the occlusion handling provides accurate dynamic/static separation, while 2D covariance invariance ensures that initial world Gaussians have reasonable sizes. Without the former, some static contents are mistakenly treated as dynamic, leading to PSNR performance drop of 0.3dB (Table 3(b)) and blurry background reconstruction especially in the regions that have been occluded (Fig. 6(b)). Without the latter, world Gaussians are not initialized well in size, resulting in sub-optimal results, as indicated in Table 3(c) and Fig. 6(c).

**Effect of HDR Photometric Reprojection Loss.** To verify the efficacy of the HDR photometric reprojection loss, we experiment by removing it in the second stage. A decrease in performance can be observed in Table 3(d) and Fig. 6(d), indicating that the dense supervision from photometric reprojection loss is beneficial for refining camera poses and scene geometry.

Table 4: Ablation results about sampling interval of the cubic Hermite spline's control points on the test frames of Real-Exp-3 and Syn-Exp-3 scenes. All experiments are trained for 15K iteration. All the metrics listed here represent PSNR except HDR-TAE.

| Sample every $N_s$ frames | Real-Exp-3 | | Syn-Exp-3 | | | |
|---|---|---|---|---|---|---|
| | LDR-OE | HDR-TAE | LDR-OE | LDR-NE | HDR | HDR-TAE |
| $N_s=1$ | **27.73** | **0.067** | **34.86** | **34.65** | **37.70** | **0.057** |
| $N_s=2$ | 27.58 | **0.067** | 34.82 | 34.60 | 37.66 | 0.058 |
| $N_s=4$ (**Ours**) | 27.65 | **0.067** | 34.75 | 34.54 | 37.64 | **0.057** |
| $N_s=8$ | 27.26 | 0.068 | 34.16 | 33.95 | 37.03 | 0.058 |
| $N_s=16$ | 26.75 | 0.068 | 33.67 | 33.47 | 36.34 | 0.058 |

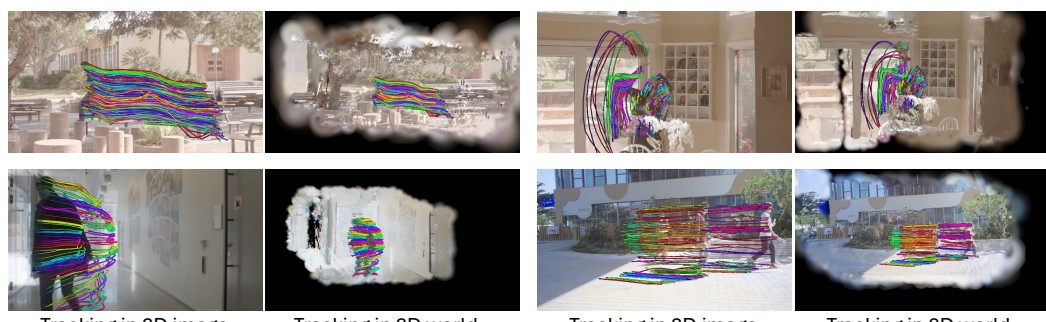

| Tracking in 2D image | Tracking in 3D world | Tracking in 2D image | Tracking in 3D world |

Figure 7: Visualization of induced 2D/3D tracking by dynamic Gaussian motion.

**Effect of Temporal Luminance Regularization.** We also investigate the influence of the temporal luminance regularization loss $\mathcal{L}_{\text{tlr}}$ by discarding it. We can see in Table 3(e) that although the reconstruction quality (PSNR) is not significantly affected, the temporal stability (TAE) is greatly degraded. This means that $\mathcal{L}_{\text{tlr}}$ plays a crucial role in stabilizing the appearance variations across frames, leading to more temporally coherent HDR videos. Fig. 6(e) also shows that without $\mathcal{L}_{\text{tlr}}$, the rendered result exhibits noticeable artifacts and noises in dynamic regions.

**Effect of the number of control points on the Cubic Hermite Spline Trajectory.** We additionally conduct ablation study on the number of control points for cubic Hermite spline trajectory. Since different scenes have different video lengths, we explore on the sample interval $N_s$ of control points. As listed in Table 4, the performance starts to degrade when $N_s$ is more than 4. We set $N_s = 4$ in our experiments as a trade-off between performance and storing memory, since smaller $N_s$ means more control points and thus more memory cost.

### 4.4 VISUALIZATION OF INDUCED 2D/3D TRACKING

We visualize the induced 2D/3D tracking by the dynamic Gaussian motion of Mono4DGS-HDR in Fig. 7. The 2D tracking is obtained by projecting the dynamic Gaussian trajectory to the image plane, while the 3D tracking is directly derived from the dynamic Gaussian trajectory in the world space. The results demonstrate that our method can also induce accurate 2D/3D tracking of dynamic objects in the scene. More results are provided in the supplementary videos.

## 5 CONCLUSION

We present Mono4DGS-HDR, the first system to tackle 4D HDR reconstruction from unposed monocular LDR videos with alternating exposures. Our novel two-stage optimization approach first learns a video HDR Gaussian representation in orthographic camera coordinate space, then transforms into world space and refines the world Gaussians along with camera poses. We also propose temporal luminance regularization to ensure the temporal consistency of HDR appearance. Experiments on our new benchmark show superior results over adapted methods in both rendering quality and speed.

ACKNOWLEDGEMENTS

The research is supported in part by Early Career Scheme of the Research Grants Council (RGC) of the Hong Kong SAR under grant No. 26202321, Department of Science & Technology of Shandong Province under grant No. SDST26EG01, SAIL Research Project, HKUST-Zeekr Coolaborative Research Fund, HKUST-WeBank Joint Lab Project, and Tencent Rhino-Bird Focused Research Program.

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

## A APPENDIX

### A.1 ADDITIONAL METHOD DETAILS

#### A.1.1 3D GAUSSIAN SPLATTING

3DGS (Kerbl et al., 2023) represents a scene using a set of anisotropic 3D Gaussians, each parameterized by its position $\mu \in \mathbb{R}^3$, rotation quaternion $r \in \mathbb{R}^4$, scaling $s \in \mathbb{R}^3$, opacity $\alpha \in [0, 1]$, and color $c \in [0, 1]^3$. The spatial distribution of each Gaussian follows $G(x) = e^{-\frac{1}{2}(x-\mu)^\top \Sigma^{-1}(x-\mu)}$, where $x$ is an arbitrary 3D position and $\Sigma = RSS^\top R^\top$ is the covariance matrix derived from the rotation matrix $R$ (or $r$) and the scaling matrix $S$ (or $s$). Each 3D Gaussian $G(x)$ is first

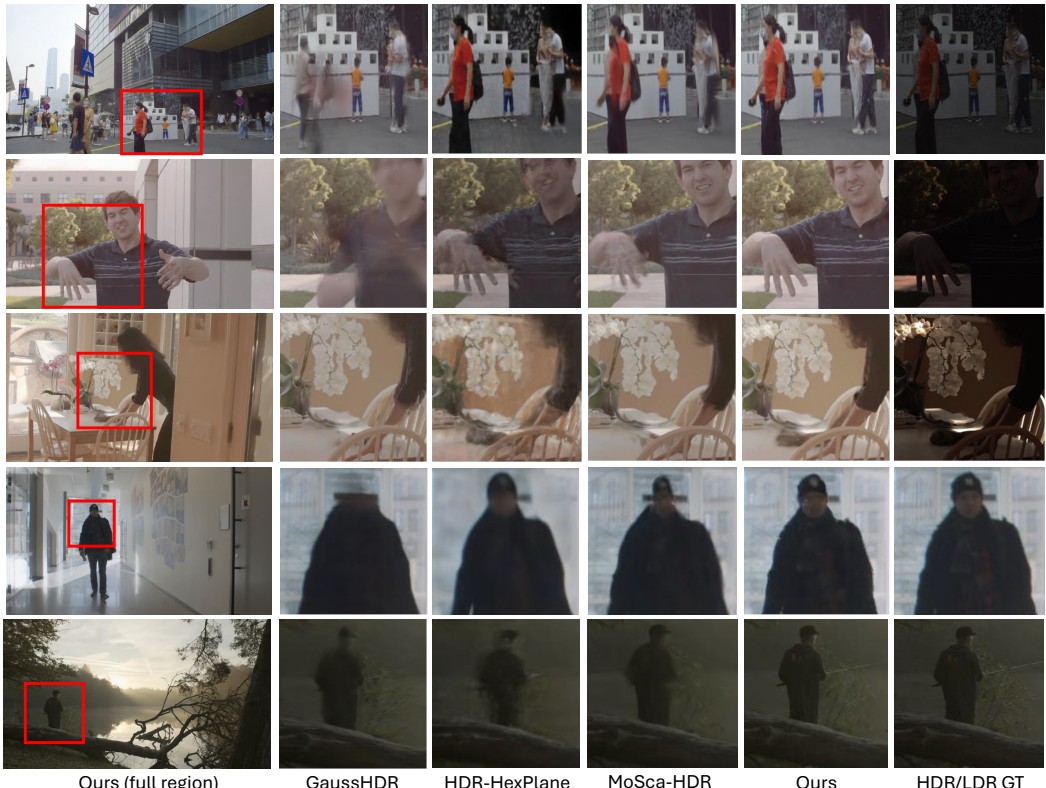

| Ours (full region) | GaussHDR | HDR-HexPlane | MoSca-HDR | Ours | HDR/LDR GT |

Figure 8: More HDR visual comparisons on train/test frames. Our method achieves superior quality.

transformed into a 2D Gaussian $G'(x)$ on the image plane by evaluating the 2D covariance $\Sigma' = [JW\Sigma W^\top J^\top]_{2\times 2}$, where $J$ is the Jacobian of the affine approximation of the projective transformation, $W$ is the viewing transformation matrix and $[\cdot]_{2\times 2}$ means skipping the third row and column. Then, a tile-based rasterizer sorts the 2D Gaussians and performs $\alpha$-blending:

$$C(u) = \sum_{i=1}^{M} c_i \sigma_i \prod_{j=1}^{i-1} (1 - \sigma_j), \quad \sigma_i = \alpha_i G'_i(u), \tag{4}$$

where $u$ denotes the queried pixel position and $M$ is the number of sorted 2D Gaussians related to the queried pixel.

### A.1.2 CUBIC HERMITE SPLINE INTERPOLATION

We use cubic Hermite spline function to model the motion of dynamic Gaussian. Given $N_c$ control points $\{p_k\}_{k=1}^{N_c}$ and their corresponding timestamps $\{t_k\}_{k=1}^{N_c}$ for cubic Hermite spline, the position of a Gaussian at time $t \in [t_k, t_{k+1}]$ can be computed as:

$$\mu(t) = h_{00}(\bar{t})p_k + h_{10}(\bar{t})(t_{k+1} - t_k)m_k + h_{01}(\bar{t})p_{k+1} + h_{11}(\bar{t})(t_{k+1} - t_k)m_{k+1},$$
$$\bar{t} = \frac{t - t_k}{t_{k+1} - t_k}, \quad m_k = \frac{1}{2}\left(\frac{p_{k+1} - p_k}{t_{k+1} - t_k} + \frac{p_k - p_{k-1}}{t_k - t_{k-1}}\right), \tag{5}$$

where $h_{00}(\bar{t}) = 2\bar{t}^3 - 3\bar{t}^2 + 1$, $h_{10}(\bar{t}) = \bar{t}^3 - 2\bar{t}^2 + \bar{t}$, $h_{01}(\bar{t}) = -2\bar{t}^3 + 3\bar{t}^2$, and $h_{11}(\bar{t}) = \bar{t}^3 - \bar{t}^2$ are the Hermite basis functions, and $m_k$ is the approximated tangent at control point $p_k$.

### A.1.3 VIDEO GAUSSIAN REPRESENTATION

Video Gaussian representation optimizes in orthographic camera coordinate space where the video's width, height and depth correspond to the $X$, $Y$ and $Z$ axes, respectively. In this space, we should use an orthographic camera model for rasterization and replace the original projection Jacobian $J$ in 3DGS with $J_{\text{ortho}} = \begin{bmatrix} w/2 & 0 & 0 \\ 0 & h/2 & 0 \end{bmatrix}$, where $w$ and $h$ are the image resolution. For video

Table 5: Quantitative comparisons with GFlow (Wang et al., 2025b) on the train frames of Real-Exp-3 and Syn-Exp-3 scenes. Metrics are averaged over all scenes. OE denotes observed-exposure results. ‡ We extend GFlow to HDR mode for fair comparison.

| Method | Real-Exp-3 (train frames) | | | | Syn-Exp-3 (train frames) | | | | | | |
| | LDR-OE | | | HDR-TAE↓ | LDR-OE | | | HDR | | | HDR-TAE↓ |
| | PSNR↑ | SSIM↑ | LPIPS↓ | | PSNR↑ | SSIM↑ | LPIPS↓ | PSNR↑ | SSIM↑ | LPIPS↓ | |
| GFlow-HDR‡ | 24.88 | 0.894 | 0.139 | 0.770 | 30.08 | 0.912 | 0.115 | 18.70 | 0.674 | 0.206 | 0.815 |
| **Mono4DGS-HDR (Ours)** | **30.09** | **0.913** | **0.067** | **0.067** | **35.18** | **0.908** | **0.082** | **37.97** | **0.960** | **0.040** | **0.057** |

Gaussian initialization, we can utilize the precomputed tracking and depth priors. Each tracklet will correspond to a video Gaussian. Consider a 2D track $\mathcal{T} = \{\tau_t | \tau_t \in \mathbb{R}^2\}_{t=1}^{N_f}$, with prior video depths $\mathcal{D} = \{D_t\}_{t=1}^{N_f}$, we can lift it to camera coordinate space as a video Gaussian trajectory $\mathcal{G}^v = \{\mu_t^v | \mu_t^v = [x_t^v, y_t^v, z_t^v]\}_{t=1}^{N_f}$, where $(x_t^v, y_t^v)$ are the normalized pixel coordinates of $\tau_t$, and $z_t^v = D_t(\tau_t)$ is the depth value at pixel $\tau_t$. Then, we sample $N_c$ initial control points from $\mathcal{G}^v$ for the spline trajectory of this dynamic video Gaussian. For Gaussian rotation quaternion, we simply initialize all polynomial coefficients with $[1, 0, 0, 0]$.

### A.1.4 SUPERVISION FROM 2D PRIORS

**LDR RGB Loss.** Following 3DGS (Kerbl et al., 2023), we use a combination of DSSIM (Wang et al., 2004) and L1 losses to compute the image reconstruction loss between LDR rendering $\widetilde{L}_t$ and GT image $L_t$, denoted as $\mathcal{L}_{\text{rgb}} = \lambda_{\text{d}} \text{DSSIM}(\widetilde{L}_t, L_t) + (1 - \lambda_{\text{d}}) \|\widetilde{L}_t - L_t\|_1$, where $\lambda_{\text{d}} = 0.2$.

**Depth Loss.** We include a depth loss on the rendered depth map $\widetilde{D}_t$ and prior depth map $D_t$, which can be formulated as $\mathcal{L}_{\text{dep}} = \|\widetilde{D}_t - D_t\|_1$.

**Flow/Track Loss.** To distill motion information from 2D priors to 3D Gaussians, we compute the flow/track loss (Lei et al., 2025; Wang et al., 2025a) between the rendered flow/track map $\widetilde{F}_{t \to t'}$ and the prior flow/track map $F_{t \to t'}$, derived as $\mathcal{L}_{\text{track}} = V_{t \to t'} \odot \|\widetilde{F}_{t \to t'} - F_{t \to t'}\|_1$, where $V_{t \to t'}$ is the valid flow mask or sparse track mask. We randomly choose to supervise with flow or track loss at each iteration. For flow supervision, we set $t' \in \{t \pm N_e\}$ where $N_e$ is the number of exposure levels (note that we only extract prior optical flow between two frames at the same exposure level). For track supervision, we randomly sample $t'$ from the entire video sequence (except $t$).

**Unit Exposure Loss.** We follow HDR-NeRF (Huang et al., 2022) to incorporate a unit exposure loss on the logarithmic-domain tone mapper $\phi$ to fix the scale of learned HDR irradiance, enabling HDR quality evaluation, expressed as $\mathcal{L}_{\text{ue}} = \|\phi(0) - C_0\|_2^2$, where $C_0 = 0.5$ for real scenes and $C_0 = 0.73$ (derived from GT CRF) for synthetic scenes.

### A.1.5 GAUSSIAN MOTION REGULARIZATION

**Rigidity Constraint.** We enforce the as-rigid-as-possible (or distance preserving) loss (Wang et al., 2025a; Sun et al., 2024) between dynamic Gaussians and their $k$-nearest neighbors to ensure the local rigidity. For each Gaussian, let $\mu_t$ and $\mu_{t'}$ be its positions at time $t$ and $t'$, and $\mathcal{N}_k(\mu_t)$ denote the set of $k$-nearest neighbors of $\mu_t$, then this loss can be defined as:

$$\mathcal{L}_{\text{arap}} = \|\text{dist}(\mu_t, \mathcal{N}_k(\mu_t)) - \text{dist}(\mu_{t'}, \mathcal{N}_k(\mu_{t'}))\|_2^2, \tag{6}$$

where $\text{dist}(\cdot, \cdot)$ measures Euclidean distance. We set $k = 5$ in our experiments.

**Motion Smoothness.** We apply the velocity and acceleration smoothness losses $\mathcal{L}_{\text{vel}}$ and $\mathcal{L}_{\text{acc}}$ to encourage smooth motion of dynamic Gaussians. Considering a dynamic Gaussian with position $\mu_t$ and rotation matrix $R_t$ at time $t$, we can follow MoSca (Lei et al., 2025) to obtain $\mathcal{L}_{\text{vel}}$ and $\mathcal{L}_{\text{acc}}$ as:

$$\mathcal{L}_{\text{vel}} = \sum_{t=1}^{N_f-1} \|\mu_{t+1} - \mu_t\|_2 + \|\log(R_t R_{t+1}^{-1})\|_F,$$

$$\mathcal{L}_{\text{acc}} = \sum_{t=1}^{N_f-2} \|(\mu_t - 2\mu_{t+1} + \mu_{t+2})\|_2 + \left| \|\log(R_t R_{t+1}^{-1})\|_F - \|\log(R_{t+1} R_{t+2}^{-1})\|_F \right|, \tag{7}$$

where $\|\log(\cdot)\|_F$ is the Frobenius norm of rotation logarithm (the axis-angle of the rotation).

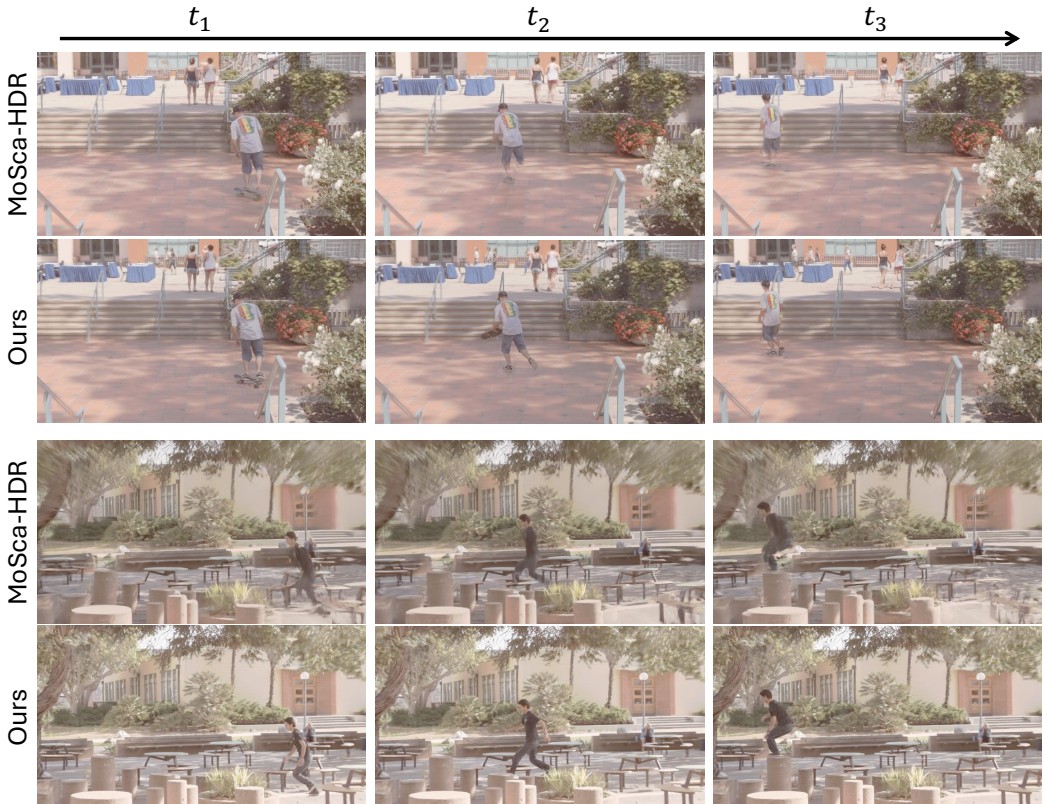

Figure 9: HDR visual comparisons under fix-view-change-time setting.

### A.1.6 HDR PHOTOMETRIC REPROJECTION LOSS

The photometric reprojection loss is widely used in self-supervised monocular depth estimation (Godard et al., 2019) to optimize depths and camera poses together. Given the query timestamp $t$ and a reference timestamp $t'$, we can compute the reference pixel coordinate $u_{t\to t'}$ corresponding to the query pixel coordinate $u_t$ as:

$$u_{t\to t'} = \pi_{\hat{K}}(\hat{R}_{t'}\hat{R}_t^{-1}(\pi_{\hat{K}}^{-1}(u_t, D_t(u_t)) - \hat{T}_t) + \hat{T}_{t'}), \tag{8}$$

where $\hat{K}$ is the camera intrinsics, $[\hat{R}_t|\hat{T}_t]$ and $[\hat{R}_{t'}|\hat{T}_{t'}]$ are the camera extrinsics at time $t$ and $t'$, $D_t(u_t)$ is the depth value at pixel $u_t$, $\pi_{\hat{K}}(\cdot)$ and $\pi_{\hat{K}}^{-1}(\cdot)$ are the projection and unprojection functions. Then we sample the corresponding HDR color of $\widetilde{H}_t(u_t)$ in the reference frame by bilinear interpolation, denoted as $\widetilde{H}_{t'}(u_{t\to t'})$. In this way, we can obtain the warped HDR image $\widetilde{H}_{t'\to t}$ from $t'$ to $t$ and compute the HDR photometric reprojection loss as:

$$\mathcal{L}_{\mathrm{pr}} = \left| (1 - M_t) \odot \frac{\widetilde{H}_{t'\to t} - \widetilde{H}_t}{\widetilde{H}_{t'\to t} + \widetilde{H}_t} \right|_1, \tag{9}$$

where $M_t$ is the mask indicating dynamic pixels and the $\widetilde{H}_{t'\to t} + \widetilde{H}_t$ normalization is to eliminate the influence of the scale of HDR values. We sample $t'$ from $\{t\pm 1, t\pm 2\}$ in the experiments. Note that we use the recovered HDR training frames in the first stage for the computation of $\mathcal{L}_{\mathrm{pr}}$ in the second stage, where we stop gradients of the leveraged HDR frames.

## A.2 ADDITIONAL EXPERIMENTAL SETTINGS

### A.2.1 DATASETS

**Syn-Exp-3.** Syn-Exp-3 contains 9 synthetic HDR videos. Each video has nearly 100 frames of resolution $864 \times 480$. We adopt an alternate frame sampling strategy, where we extract every other

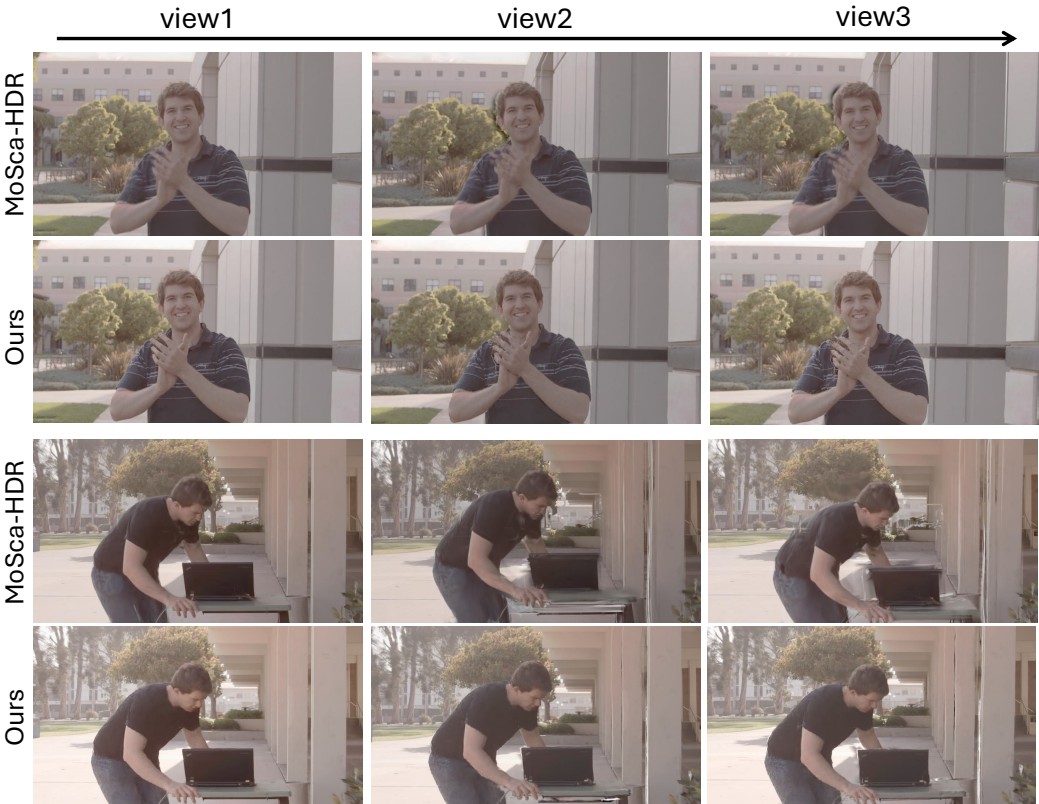

Figure 10: HDR visual comparisons under fix-time-change-view setting.

frame from the original sequence. Frames with odd indices are used for training, while frames with even indices are reserved for testing, resulting in an equal 50% split between training and testing data. The test frames are only used for quantitative evaluation and not for training, which are at novel (interpolated) views and times compared to training frames. Since the GT HDR videos are available, we can use them to create alternating-exposure LDR inputs. Specifically, we leverage a predefined CRF function as in HDR-NeRF (Huang et al., 2022) to tone-map the HDR frame $H$ to LDR frame $L$ with a given exposure time $\Delta t$, which is formulated as:

$$L = f(H\Delta t) = \left(\frac{H\Delta t}{H\Delta t + 1}\right)^{\frac{1}{2.2}}, \tag{10}$$

where we can derive the unit exposure loss GT as $C_0 = f(1) = 0.73$. In this way, we generate LDR frames with 3 (observed) exposure levels for training (LDR-OE). For test frames, we additionally create LDR images at another 2 exposure levels for novel exposure evaluation (LDR-NE).

**Real-Exp-3.** Real-Exp-3 contains 8 real-world alternating-exposure LDR videos with 3 exposure levels. Each video has nearly 50-60 frames of resolution $864 \times 480$. We follow the same alternate frame sampling strategy as Syn-Exp-3 to split training and testing frames. The LDR inputs are directly taken from the original sequences. For testing frames, we can only evaluate on the observed exposures (LDR-OE).

**Real-Exp-2.** Real-Exp-2 contains 8 real-world alternating-exposure LDR videos with 2 exposure levels. Each video has nearly 50-60 frames of resolution $864 \times 480$. Since there are only 2 exposures, it is inconvenient to split training and testing frames using the alternate frame sampling strategy. Therefore, we directly apply the whole video sequence for training and evaluate on all training frames at the observed exposures (LDR-OE).

### A.2.2 EVALUATION METRICS

**HDR-TAE.** Similar to the temporal alignment error (TAE) in video depth assessment (Yang et al., 2025), we introduce HDR-TAE to measure the temporal consistency of rendered HDR videos in

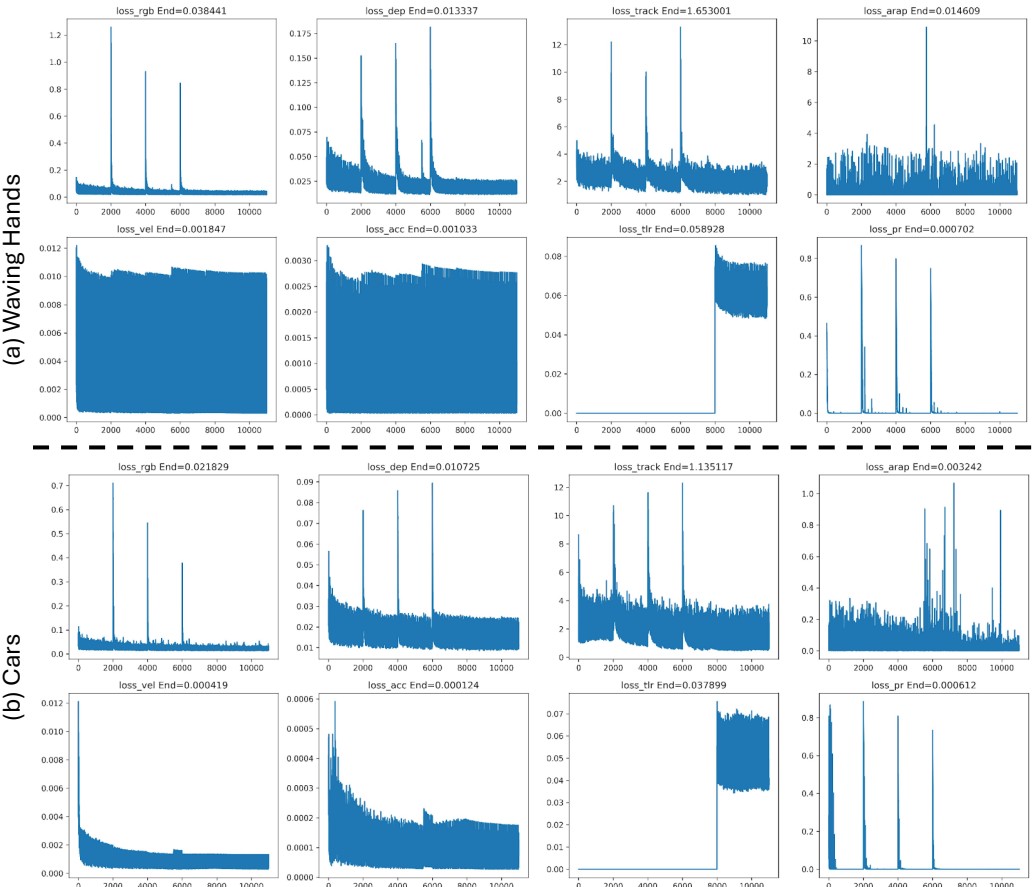

Figure 11: The core loss curves on two scenes: WavingHands and Cars.

per-pixel manner, defined as:

$$\text{HDR-TAE} = \frac{1}{2(N_f - 1)} \sum_{t=1}^{N_f - 1} \left| V_{t \rightarrow t+1} \odot \frac{\widetilde{H}_{t+1 \rightarrow t} - \widetilde{H}_t}{\widetilde{H}_{t+1 \rightarrow t} + \widetilde{H}_t} \right|_1 + \left| V_{t+1 \rightarrow t} \odot \frac{\widetilde{H}_{t \rightarrow t+1} - \widetilde{H}_{t+1}}{\widetilde{H}_{t \rightarrow t+1} + \widetilde{H}_{t+1}} \right|_1,$$
$$\widetilde{H}_{t+1 \rightarrow t} = \mathcal{W}(\widetilde{H}_{t+1}, F_{t \rightarrow t+1}), \quad \widetilde{H}_{t \rightarrow t+1} = \mathcal{W}(\widetilde{H}_t, F_{t+1 \rightarrow t}),$$

(11)

where $\mathcal{W}(\cdot, \cdot)$ means backward warping. The optical flows $\{F_{t \rightarrow t+1}, F_{t+1 \rightarrow t}\}$ and visibility masks $\{V_{t \rightarrow t+1}, V_{t+1 \rightarrow t}\}$ are extracted between the adjacent HDR frames $\widetilde{H}_t$ and $\widetilde{H}_{t+1}$ (tone-mapped version) using RAFT (Teed & Deng, 2020) model. For real-world scenes, we utilize the rendered HDR video frames. For synthetic scenes, we directly leverage the GT HDR video frames. Lower HDR-TAE indicates better temporal consistency.

### A.2.3 IMPLEMENTATION DETAILS

We process on the original resolution of $864 \times 480$ for all videos. We threshold the epipolar error maps by 0.00001 to obtain the dynamic masks. For the cubic Hermite spline trajectory of each dynamic Gaussian, we sample the control points every 4 frames. For HDR Gaussian color, we store 36-dimensional features and compute HDR color via a color MLP instead of spherical harmonics in 3DGS for stability. The color and tone-mapper MLPs consist of one hidden layer with 36 and 128 channels respectively. We follow previous works (Huang et al., 2022; Cai et al., 2024; Liu et al., 2025a) to use three different MLPs to model RGB-channel tone mappers independently. Besides, we simply adopt 2D tone mapping described in GaussHDR (Liu et al., 2025a) for stable HDR reconstruction, where we first render the HDR Gaussians to an HDR image and then tone-map it to an LDR image. The loss weights are set as $\lambda_{\text{rgb}} = 1$, $\lambda_{\text{ue}} = 10$, $\lambda_{\text{dep}} = 1$, $\lambda_{\text{track}} = 0.01$, $\lambda_{\text{arap}} = 0.01$, $\lambda_{\text{vel}} = 10$, $\lambda_{\text{acc}} = 10$, $\lambda_{\text{tlr}} = 0.1$ and $\lambda_{\text{pr}} = 1$. We optimize the first stage (fully dynamic video

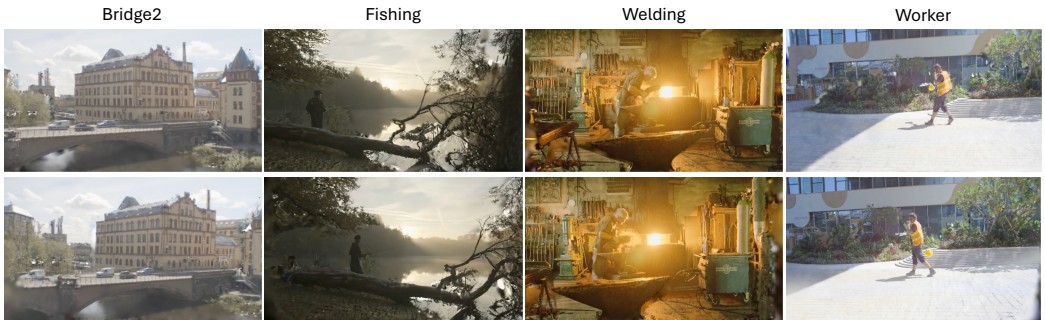

Figure 12: Wide-view-change rendering results on four scenes.

Table 6: Ablation results about other exposure schedules on the test frames of Syn-Exp-3 scenes. Metrics are averaged over all scenes. LDR-OE and LDR-NE denote the LDR results with observed and novel exposures, respectively. HDR denotes the HDR results.

| Method | LDR-OE | | | LDR-NE | | | HDR | | | |
|---|---|---|---|---|---|---|---|---|---|---|
| | PSNR↑ | SSIM↑ | LPIPS↓ | PSNR↑ | SSIM↑ | LPIPS↓ | PSNR↑ | SSIM↑ | LPIPS↓ | TAE↓ |
| random 3-exposure schedule | 34.71 | 0.902 | 0.093 | 34.52 | 0.914 | 0.087 | 37.37 | 0.956 | 0.052 | **0.057** |
| random exposure values | 34.59 | 0.902 | 0.094 | 34.44 | 0.913 | 0.088 | 37.42 | 0.955 | 0.053 | **0.057** |
| **Ours** | **34.75** | **0.904** | **0.086** | **34.54** | **0.915** | **0.081** | **37.64** | **0.959** | **0.042** | **0.057** |

Gaussians) for 4K iterations, with 3K iterations of Gaussian densification where we densify every 200 steps and reset opacities every 1K steps. The second stage (static and dynamic world Gaussians) is optimized for 11K iterations with 8K iterations of Gaussian densification. For static Gaussians, we densify every 400 steps and reset opacities every 2K steps. For dynamic Gaussians, we densify every 200 steps, reset opacities every 2K steps and remove mistakenly densified Gaussians every 2K steps. In both stages, the temporal luminance regularization loss $\mathcal{L}_{\text{tlr}}$ is applied after the end of Gaussian densification. The learning rates of static Gaussian attributes are same as 3DGS (Kerbl et al., 2023). The learning rates of dynamic Gaussian attributes and camera parameters are following MoSca (Lei et al., 2025). The learning rate of tone-mapper and color MLPs is set to 0.0005. All experiments are conducted with Adam optimizer (Kingma & Ba, 2015) using Pytorch (Paszke et al., 2019) on a single RTX 3090 GPU.

### A.3 ADDITIONAL EXPERIMENTAL RESULTS

#### A.3.1 PERFORMANCE COMPARISONS

We provide more comparison results with GFlow (Wang et al., 2025b) on training frames of Real-Exp-3 and Syn-Exp-3 scenes in Table 5. We also present more HDR visual comparisons on train/test frames in Fig. 8, and the results under fix-view-change-time and fix-time-change-view settings in Fig. 9 and Fig. 10, respectively. All these results demonstrate again the superiority of our method.

#### A.3.2 LOSS CURVES

We present the loss curves for two scenes, WavingHands and Cars, as illustrated in Fig. 11. Due to the resetting of Gaussian opacity, certain losses, such as RGB loss, depth loss, and track loss, may exhibit sudden increases at the resetting iterations. For the TLR loss, it is applied only after the completion of Gaussian densification, so it appears during the later training stage. Despite the presence of multiple losses, they function collaboratively and converge effectively. The stability of the optimization process is attributed to several factors. First, we utilize precomputed 2D priors to guide the optimization, providing effective initializations and constraints for geometry and motion. Second, we implement a two-stage optimization strategy with video Gaussian initialization, which enhances the stability and efficiency of world Gaussian optimization.

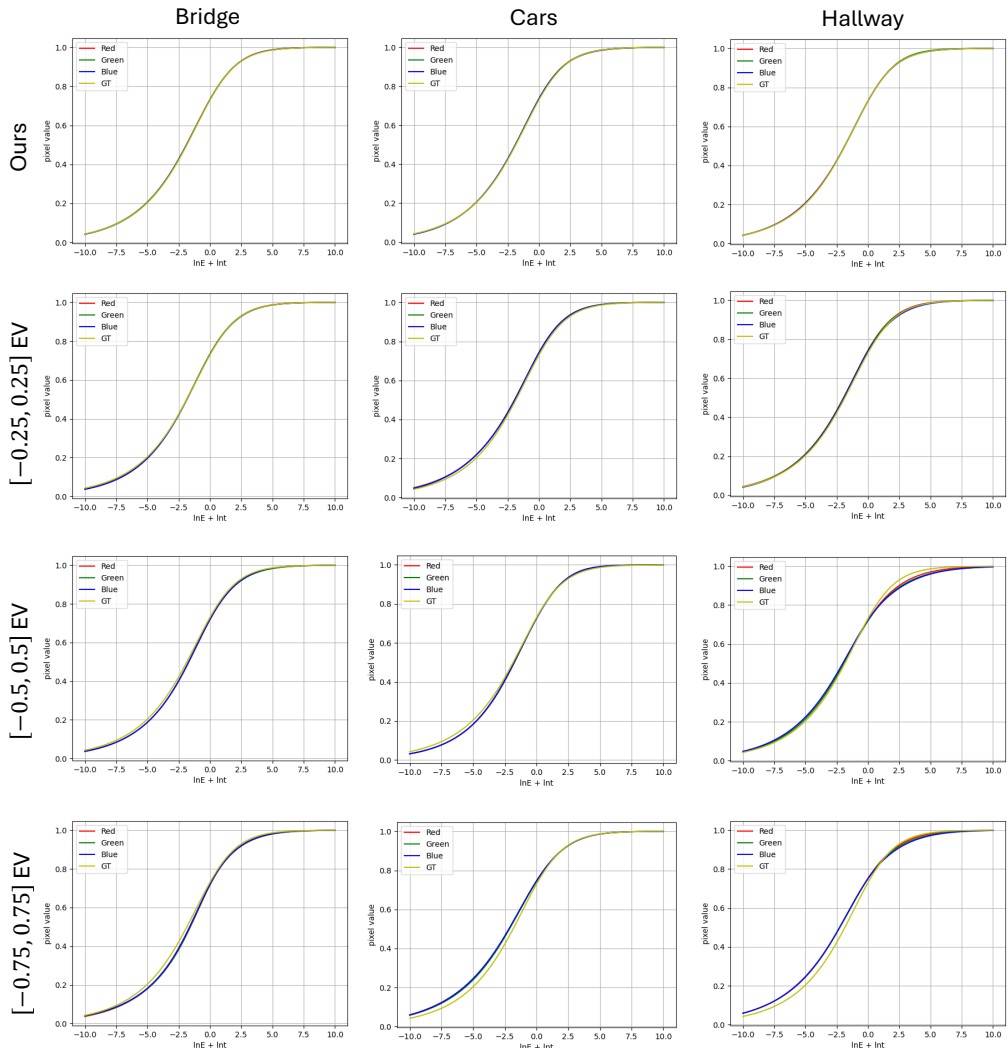

Figure 13: Recovered CRFs under different random exposure perturbation ranges.

### A.3.3 WIDE-VIEW-CHANGE RENDERING

As we do not possess ground truth HDR/LDR images under significant view changes, we are unable to quantitatively evaluate performance. However, we present visual results under wide view changes in Fig. 12 for four scenes. Please refer to our supplementary videos for more results. Although some artifacts are present, our method maintains reasonable geometric and photometric consistency under wide view changes.

### A.3.4 MORE ABLATION RESULTS

**Other Exposure Schedules.** In our work, we primarily utilize alternating-exposure video data, as this type of data is convenient to capture and commonly employed in existing HDR video reconstruction studies (Xu et al., 2024; Chung & Cho, 2023). Additionally, we explore other exposure schedules, including a random (non-periodic) 3-exposure schedule and random exposure values. To facilitate this, we regenerate the training LDR frames from the HDR ground truths in the Syn-Exp-3 scenes. For the random 3-exposure schedule, we retain the original three exposure values but randomly shuffle their order within the training video sequence. For random exposure values, we assign a new exposure value to each training frame at random. In both cases, we utilize the original exposure values for the testing frames. The results are summarized in Table 6, where we observe that our method continues to perform well under these exposure schedules. The PSNR and SSIM metrics

Table 7: Ablation results about different random exposure perturbation ranges on the test frames of Syn-Exp-3 scenes. Metrics are averaged over all scenes. LDR-OE and LDR-NE denote the LDR results with observed and novel exposures, respectively. HDR denotes the HDR results.

| Random Perturbation | LDR-OE | | | LDR-NE | | | HDR | | | |
|---|---|---|---|---|---|---|---|---|---|---|
| | PSNR↑ | SSIM↑ | LPIPS↓ | PSNR↑ | SSIM↑ | LPIPS↓ | PSNR↑ | SSIM↑ | LPIPS↓ | TAE↓ |
| $[-0.75, 0.75]$ EV | 31.37 | 0.896 | 0.094 | 31.32 | 0.908 | 0.089 | 34.32 | 0.953 | 0.048 | 0.060 |
| $[-0.5, 0.5]$ EV | 33.31 | 0.901 | 0.091 | 33.21 | 0.912 | 0.086 | 35.55 | 0.957 | 0.046 | 0.058 |
| $[-0.25, 0.25]$ EV | 34.27 | 0.902 | 0.083 | 34.13 | 0.913 | 0.083 | 37.28 | 0.958 | 0.044 | 0.058 |
| **Ours** | **34.75** | **0.904** | **0.086** | **34.54** | **0.915** | **0.081** | **37.64** | **0.959** | **0.042** | **0.057** |

Table 8: Ablation results about depth and flow/track losses on the test frames of Syn-Exp-3 scenes. Metrics are averaged over all scenes. LDR-OE and LDR-NE denote the LDR results with observed and novel exposures, respectively. HDR denotes the HDR results.

| Method | LDR-OE | | | LDR-NE | | | HDR | | | |
|---|---|---|---|---|---|---|---|---|---|---|
| | PSNR↑ | SSIM↑ | LPIPS↓ | PSNR↑ | SSIM↑ | LPIPS↓ | PSNR↑ | SSIM↑ | LPIPS↓ | TAE↓ |
| w/o depth loss | 33.63 | 0.897 | 0.095 | 33.55 | 0.908 | 0.090 | 35.56 | 0.953 | 0.054 | 0.058 |
| w/o track/flow loss | 33.43 | 0.896 | 0.097 | 33.23 | 0.909 | 0.092 | 35.05 | 0.952 | 0.052 | 0.058 |
| **Ours** | **34.75** | **0.904** | **0.086** | **34.54** | **0.915** | **0.081** | **37.64** | **0.959** | **0.042** | **0.057** |

exhibit only slight reductions, while the drop in the LPIPS metric is more pronounced. This larger decline can be attributed to less accurate optical flow estimation under these exposure schedules, as the flow model is applied to two frames with differing exposures. Although the flow model demonstrates some robustness to exposure changes due to brightness augmentation during its pretraining, it may still encounter difficulties with substantial exposure differences, leading to less accurate flow priors and, consequently, poorer LPIPS results.

**Sensitivity to Inaccuracies of Exposure Values.** In our work, we use the known exposure values read from camera metadata and treat the camera response function (CRF) as unknown and optimizable. As depicted in Fig. 13, our learned tone mappers can well approximate the ground truth CRF curves with knowing the exact exposure values. Here, we investigate the sensitivity of our method to the inaccuracies of exposure values. Specifically, we randomly perturb the input exposure values with different perturbation ranges during training. As listed in Table 7, our method is robust to small exposure perturbations (e.g., $[-0.25, 0.25]$ EV) with a slight performance drop. However, larger perturbations lead to more significant performance degradation, which aligns with intuitive expectations. We can also see the recovered CRF curves under different perturbation ranges in Fig. 13, where larger exposure perturbations result in poorer CRF fitting. Notably, exposure values can typically be accurately extracted from camera metadata in real-world applications, affirming the practicality of our method.

**Influence of Depth/Flow/Track Priors.** The depth/flow/track priors are extracted by vision foundation models, which may contain some noise inherently. Here, we also explore the performance of our method when these priors are absent, i.e., without depth loss or flow/track loss. We provide the ablation results on Syn-Exp-3 scenes in Table 8. We can see that the performance drops a lot without these losses, demonstrating the importance of the priors for high-quality reconstruction.

## A.4 CHALLENGING CONDITIONS

Our experimental datasets encompass challenging scenarios, including scenes with low-light conditions, non-Lambertian (reflective/transparent) surfaces, and fast or complex motions, as illustrated in Fig. 14. Some of these scenes are also presented in our supplementary videos. Although our method is designed for general HDR reconstruction from varying-exposure LDR frames and does not specifically address these difficult conditions, it still produces reasonable results. To further enhance performance in such challenging settings, we can incorporate specialized techniques from existing literature.

In low-light scenes, such as those depicted in Fig. 14(a)(b), which are often characterized by inherent noise, operating in the RAW domain offers significant advantages. This is evidenced by studies such as RawNeRF (Mildenhall et al., 2022) and LE3D (Jin et al., 2024), which are specifically designed for low-light or dark environments. While our current framework processes alternating-

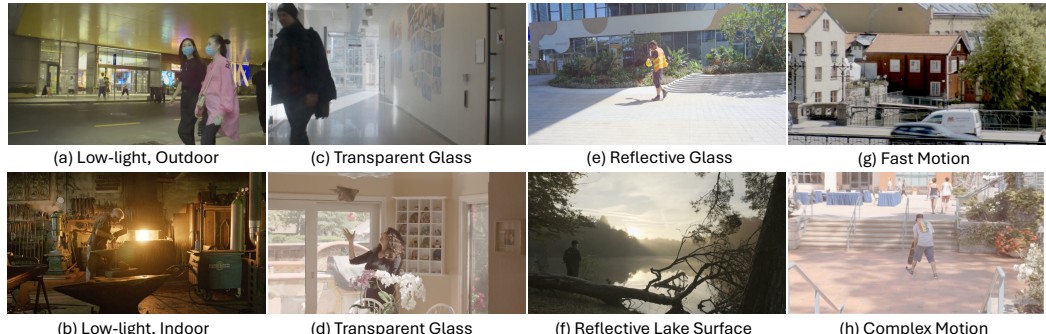

Figure 14: Several challenging conditions that are involved in our experimental datasets.

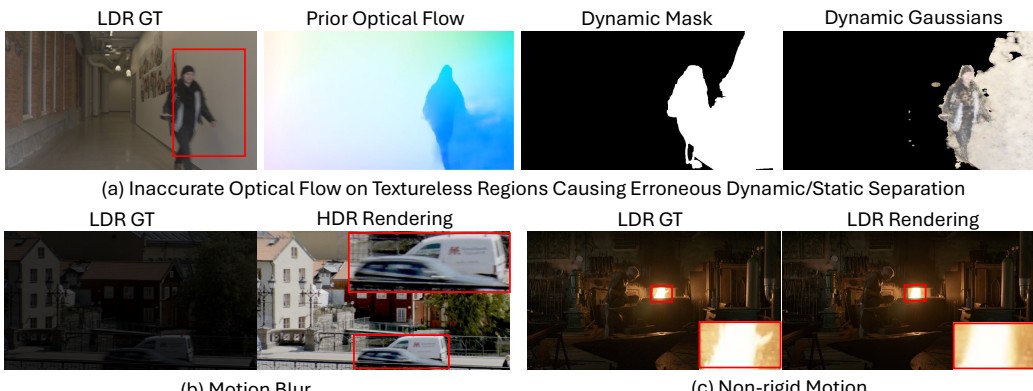

Figure 15: Several typical failure cases of Mono4DGS-HDR.

exposure videos, we can easily combine our underlying scene representation with RAW-domain color representation to better manage low-light scenarios.

For transparent surfaces in Fig. 14(c)(d) and reflective surfaces in Fig. 14(e)(f), we can integrate reflection/refraction modeling as in GaussianShader (Jiang et al., 2024) and TransparentGS (Huang et al., 2025). Since our framework adopts Gaussian Splatting as the scene representation, such modeling can be readily incorporated to improve the handling of these complex surface properties.

In the fast-motion scene depicted in Fig. 14(g), the captured LDR frames often exhibit motion blur. Consequently, regions affected by blur tend to yield similarly blurry reconstructions due to the pixel-level photometric supervision employed. Nevertheless, the overall reconstruction quality remains acceptable. Although deblurring is not the primary focus of our approach, deblurring techniques such as Deblur4DGS (Wu et al., 2024d) could be seamlessly integrated to enable simultaneous deblurring and HDR reconstruction.

We also include a scene, Skateboarder, with complex motion, as shown in Fig. 14(h). This scene contains several types of motion, including a skateboarder moving rapidly nearby, pedestrians walking slowly in the distance, and flowers swaying in the wind. Additional results for this scene are available in our supplementary videos, which demonstrate that our method effectively manages complex motion. This capability can be attributed to our dynamic Gaussian representation and video Gaussian initialization. The former enables each dynamic Gaussian to possess its own motion trajectory, thereby effectively modeling diverse motion patterns within the scene. The latter facilitates the motion learning of dynamic Gaussians by mitigating the interference of camera motion during the video Gaussian stage.

## A.5 LIMITATIONS

Although our method achieves superior performance on monocular 4D HDR reconstruction, it still has some limitations. First, our method relies on the quality of 2D priors (depth, flow and track)

to a certain extent. Inaccurate priors may lead to suboptimal results. For example, erroneous depth priors can cause incorrect scene geometry, while inaccurate flow/track priors can misguide the motion of dynamic Gaussians. Besides, the dynamic masks (epipolar error maps) derived from optical flow priors may also not be perfect, which can affect the separation of static and dynamic world Gaussians. Second, our method cannot handle the image blur caused by fast camera/object motion. Luckily, we observe that coping with blurry monocular videos in Gaussian splatting has been discussed in Deblur4DGS (Wu et al., 2024d) and Casual3DHDR (Gong et al., 2025), which provide possibilities to achieve deblurring and HDR reconstruction simultaneously. Third, Our method fails to model the complex non-rigid motion due to the as-rigid-as-possible (ARAP) constraint, where we do not consider non-rigid deformation of dynamic objects.

We also present qualitative examples in Fig. 15 that illustrate the typical failure cases mentioned above, including: (a) erroneous dynamic/static separation due to inaccurate optical flow estimation; (b) motion blur resulting from rapid camera or object motion; and (c) non-rigid deformation of burning fire. Although these limitations exist, they do not detract from our core contribution of introducing Mono4DGS-HDR for general monocular 4D HDR reconstruction. Our method demonstrates superior performance compared to existing approaches and performs well in most scenarios, as evidenced by extensive quantitative and qualitative results. Furthermore, the identified limitations offer valuable directions for future research.

### A.6 ETHICS STATEMENT

This research methodology does not raise any ethical issues. No human subjects were involved.

### A.7 REPRODUCIBILITY STATEMENT

Our Mono4DGS-HDR is built by integrating the publicly available codebase of 3DGS (Kerbl et al., 2023), MoSca (Lei et al., 2025), SaV (Sun et al., 2024), GaussHDR (Liu et al., 2025a), Spatial-Tracker (Xiao et al., 2024), DepthCrafter (Hu et al., 2025), and RAFT (Teed & Deng, 2020). We create our evaluation benchmark based on the publicly available datasets (Kronander et al., 2014; Froehlich et al., 2014; Kalantari et al., 2013; Chen et al., 2021) for HDR video reconstruction. In this paper, we include comprehensive data preprocessing and implementation details, which greatly facilitate reproducing our work. We will release our code and processed data if this work is accepted.

### A.8 LLM USAGE

We use the LLMs including GPT-4 (OpenAI et al., 2024) to help polish the writing of the paper.

