# OpenReview forum: "Mono4DGS-HDR: High Dynamic Range 4D Gaussian Splatting from Alternating-exposure Monocular Videos"
_ICLR.cc/2026/Conference — ICLR 2026 Poster_

### Official Review · Reviewer_vqUK · 2025-10-29

**Soundness:** 3
**Presentation:** 3
**Contribution:** 4
**Rating:** 8
**Confidence:** 3

**Summary:**

This paper introduces a system for reconstructing 4D HDR scenes from alternating-exposure monocular LDR videos without known camera poses. The method builds on 3D Gaussian Splatting and proposes a two-stage optimization framework:
- A video-space stage, which learns dynamic HDR Gaussians in orthographic camera coordinates, eliminating the need for camera poses.
- A world-space stage, which transforms and refines these Gaussians jointly with camera poses using HDR photometric reprojection.
Additionally, a Temporal Luminance Regularization is introduced to ensure temporal consistency of HDR appearance.

**Strengths:**

-  it is the first to handle alternating-exposure monocular HDR reconstruction.

- Well-designed two-stage optimization, effectively bridging unposed monocular input to HDR Gaussian representation.

- Good experiment results, outperforming both 3DGS- and NeRF-based HDR methods in quality and speed, and comprehensive ablation studies, validating each component’s contribution.

**Weaknesses:**

- Dependence on multiple vision foundation models (DepthCrafter, RAFT, etc.) makes the pipeline complex and may limit real-time applicability.

- The approach assumes alternating exposure patterns; performance under random or adaptive exposure schedules is not analyzed.

- Large novel view rendering is not demonstrated. It remains unclear whether the reconstructed HDR scenes maintain geometric and photometric consistency under wide view changes

**Questions:**

- How sensitive is the method to inaccuracies in the exposure timing or camera response function estimation?

- Could the framework generalize to arbitrary (non-periodic) exposure sequences?

- How large is the performance drop if vision priors (depth/flow) are noisy or absent?

---

> ### Author Response · Authors · 2025-11-25
>
> We sincerely thank the Reviewer vqUK for the insightful comments. Please find our responses below.
>
> > ### W1. Dependence on multiple vision foundation models (DepthCrafter, RAFT, etc.) makes the pipeline complex and may limit real-time applicability.
>
> Using vision foundation models to provide motion/geometry priors is a common practice in modern 3D/4D reconstruction, especially in monocular settings, since 4D reconstruction from monocular videos is highly ill-posed. Previous works (MoSca [1], SoM [2], SplineGS [3]) have also demonstrated that these priors can significantly improve reconstruction quality by providing scene initialization and regularization. Moreover, we just perform inference on the foundation models without any fine-tuning, which does not bring too much complexity to the pipeline. The inference of foundation models is a one-time pre-processing step that accounts for a small portion of the overall training time (e.g., less than 10% in our experiments) and does not impact the real-time rendering speed.
>
> > ### W2. The approach assumes alternating exposure patterns; performance under random or adaptive exposure schedules is not analyzed.
> > ### Q2. Could the framework generalize to arbitrary (non-periodic) exposure sequences?
>
> In our work, we primarily utilize alternating-exposure video data, as this type of data is convenient to capture and commonly employed in existing HDR video reconstruction studies [4,5]. Additionally, we explore other exposure schedules, including a random (non-periodic) 3-exposure schedule and random exposure values. To facilitate this, we regenerate the training LDR frames from the HDR ground truths in the Syn-Exp-3 scenes. For the random 3-exposure schedule, we retain the original three exposure values but randomly shuffle their order within the training video sequence. For random exposure values, we assign a new exposure value to each training frame at random. In both cases, we utilize the original exposure values for the testing frames. The results are summarized in Table R3 below, where we observe that our method continues to perform well under these exposure schedules. The PSNR and SSIM metrics exhibit only slight reductions, while the drop in the LPIPS metric is more pronounced. This larger decline can be attributed to less accurate optical flow estimation under these exposure schedules, as the flow model is applied to two frames with differing exposures. Although the flow model demonstrates some robustness to exposure changes due to brightness augmentation during its pretraining, it may still encounter difficulties with substantial exposure differences, leading to less accurate flow priors and, consequently, poorer LPIPS results.
>
> **Table R3**: Ablation results about other exposure schedules on the test frames of Syn-Exp-3 scenes. Metrics are averaged over all scenes. LDR-OE and LDR-NE denote the LDR results with observed and novel exposures, respectively. HDR denotes the HDR results.
>
> | Method | LDR-OE | | | LDR-NE | | | HDR | | | |
> |:-------|:--------------:|:--------------:|:-----------------:|:--------------:|:--------------:|:-----------------:|:--------------:|:--------------:|:-----------------:|:---------------:|
> | | PSNR$\uparrow$ | SSIM$\uparrow$ | LPIPS$\downarrow$ | PSNR$\uparrow$ | SSIM$\uparrow$ | LPIPS$\downarrow$ | PSNR$\uparrow$ | SSIM$\uparrow$ | LPIPS$\downarrow$ | TAE$\downarrow$ |
> | random 3-exposure schedule | 34.71 | 0.902 | 0.093 | 34.52 | 0.914 | 0.087 | 37.37 | 0.956 | 0.052 | **0.057** |
> | random exposure values | 34.59 | 0.902 | 0.094 | 34.44 | 0.913 | 0.088 | 37.42 | 0.955 | 0.053 | **0.057** |
> | **Ours** | **34.75** | **0.904** | **0.086** | **34.54** | **0.915** | **0.081** | **37.64** | **0.959** | **0.042** | **0.057** |
>
> > ### W3. Large novel view rendering is not demonstrated. It remains unclear whether the reconstructed HDR scenes maintain geometric and photometric consistency under wide view changes.
>
> As we do not possess ground truth HDR/LDR images under significant view changes, we are unable to quantitatively evaluate performance. However, we present visual results under wide view changes in Fig. 15 for four scenes. Although some artifacts are present, our method maintains reasonable geometric and photometric consistency under wide view changes. More results can be seen in our supplementary videos. Please redownload the updated supplementary material and open the project webpage to view these results in the "Wide-View-Change Rendering" Part.

---

> > ### Author Response · Authors · 2025-11-25
> >
> > > ### Q1. How sensitive is the method to inaccuracies in the exposure timing or camera response function estimation?
> >
> > In our work, we use the known exposure values read from camera metadata and treat the camera response function (CRF) as unknown and optimizable. As depicted in Fig. 14, our learned tone mappers can well approximate the ground truth CRF curves with knowing the exact exposure values. Here, we investigate the sensitivity of our method to the inaccuracies of exposure values. Specifically, we randomly perturb the input exposure values with different perturbation ranges during training. As listed in Table R4 below, our method is robust to small exposure perturbations (e.g., $[-0.25, 0.25]$ EV) with a slight performance drop. However, larger perturbations lead to more significant performance degradation, which aligns with intuitive expectations. We can also see the recovered CRF curves under different perturbation ranges in Fig. 14, where larger exposure perturbations result in poorer CRF fitting. Notably, exposure values can typically be accurately extracted from camera metadata in real-world applications, affirming the practicality of our method.
> >
> > **Table R4**: Ablation results about different random exposure perturbation ranges on the test frames of Syn-Exp-3 scenes. Metrics are averaged over all scenes. LDR-OE and LDR-NE denote the LDR results with observed and novel exposures, respectively. HDR denotes the HDR results.
> > | Random Perturbation | LDR-OE | | | LDR-NE | | | HDR | | | |
> > |:-------------------|:--------------:|:--------------:|:-----------------:|:--------------:|:--------------:|:-----------------:|:--------------:|:--------------:|:-----------------:|:---------------:|
> > | | PSNR$\uparrow$ | SSIM$\uparrow$ | LPIPS$\downarrow$ | PSNR$\uparrow$ | SSIM$\uparrow$ | LPIPS$\downarrow$ | PSNR$\uparrow$ | SSIM$\uparrow$ | LPIPS$\downarrow$ | TAE$\downarrow$ |
> > | $[-0.75, 0.75]$ EV | 31.37 | 0.896 | 0.094 | 31.32 | 0.908 | 0.089 | 34.32 | 0.953 | 0.048 | 0.060 |
> > | $[-0.5, 0.5]$ EV | 33.31 | 0.901 | 0.091 | 33.21 | 0.912 | 0.086 | 35.55 | 0.957 | 0.046 | 0.058 |
> > | $[-0.25, 0.25]$ EV | 34.27 | 0.902 | 0.083 | 34.13 | 0.913 | 0.083 | 37.28 | 0.958 | 0.044 | 0.058 |
> > | **Ours** | **34.75** | **0.904** | **0.086** | **34.54** | **0.915** | **0.081** | **37.64** | **0.959** | **0.042** | **0.057** |
> >
> > > ### Q3. How large is the performance drop if vision priors (depth/flow) are noisy or absent?
> >
> > The depth/flow/track priors are extracted by vision foundation models, which may contain some noise inherently. Here, we also explore the performance of our method when these priors are absent, i.e., without depth loss or flow/track loss. We provide the ablation results on Syn-Exp-3 scenes in Table R5 below. We can see that the performance drops a lot without these losses, demonstrating the importance of the priors for high-quality reconstruction.
> >
> > **Table R5**: Ablation results about depth and flow/track losses on the test frames of Syn-Exp-3 scenes. Metrics are averaged over all scenes. LDR-OE and LDR-NE denote the LDR results with observed and novel exposures, respectively. HDR denotes the HDR results.
> >
> > | Method | LDR-OE | | | LDR-NE | | | HDR | | | |
> > |:-------|:--------------:|:--------------:|:-----------------:|:--------------:|:--------------:|:-----------------:|:--------------:|:--------------:|:-----------------:|:---------------:|
> > | | PSNR$\uparrow$ | SSIM$\uparrow$ | LPIPS$\downarrow$ | PSNR$\uparrow$ | SSIM$\uparrow$ | LPIPS$\downarrow$ | PSNR$\uparrow$ | SSIM$\uparrow$ | LPIPS$\downarrow$ | TAE$\downarrow$ |
> > | w/o depth loss | 33.63 | 0.897 | 0.095 | 33.55 | 0.908 | 0.090 | 35.56 | 0.953 | 0.054 | 0.058 |
> > | w/o track/flow loss | 33.43 | 0.896 | 0.097 | 33.23 | 0.909 | 0.092 | 35.05 | 0.952 | 0.052 | 0.058 |
> > | **Ours** | **34.75** | **0.904** | **0.086** | **34.54** | **0.915** | **0.081** | **37.64** | **0.959** | **0.042** | **0.057** |
> >
> > ### References
> > [1] Jiahui Lei et al., MoSca: Dynamic Gaussian Fusion from Casual Videos via 4D Motion Scaffolds, CVPR, 2025.
> >
> > [2] Qianqian Wang et al., Shape of Motion: 4D Reconstruction from a Single Video, ICCV, 2025.
> >
> > [3] Jongmin Park et al., SplineGS: Robust Motion-Adaptive Spline for Real-Time Dynamic 3D Gaussians from Monocular Video, CVPR, 2025.
> >
> > [4] Haesoo Chung and Nam Ik Cho, Lan-hdr: Luminance-based alignment network for high dynamic range video reconstruction, ICCV, 2023.
> >
> > [5] Gangwei Xu et al., HDRFlow: Real-Time HDR Video Reconstruction with Large Motions, CVPR, 2024.

---

> > > ### Comment · Reviewer_vqUK · 2025-11-27
> > >
> > > Thank you to the authors for the detailed and informative rebuttal. The additional analyses addressed my concerns effectively. The method’s robustness to small exposure perturbations and the demonstrated importance of vision priors further strengthen the technical soundness of the work. Plz include them in your final revision.
> > >
> > > Overall, I believe the paper makes a meaningful and well-substantiated contribution to HDR scene reconstruction, and the clarified experiments reinforce its practicality. I maintain my original score.
> > > Final Rating: 8 (accept, good paper, poster).

---

### Official Review · Reviewer_J4n4 · 2025-10-31

**Soundness:** 3
**Presentation:** 3
**Contribution:** 3
**Rating:** 6
**Confidence:** 4

**Summary:**

This paper proposes HDR-4DGS, a two-stage framework for reconstructing HDR scenes from monocular alternating exposure videos. Unlike previous works that handle dynamic Gaussians implicitly, this method explicitly parameterizes the motion of Gaussians, enabling more accurate transformation of video Gaussians into the world coordinate system. Additionally, various optimization losses are introduced to ensure high-quality final rendering.

**Strengths:**

1. Explicitly parameterizing the motion of Gaussians not only improves rendering quality but also maintains a relatively fast rendering speed.
2. The invariance of 2D Gaussian covariance serves as a simple yet effective tool introduced by the authors, which is validated through ablation studies in the paper.
3. The entire paper is clear and easy to understand.

**Weaknesses:**

1. The division between dynamic and static regions relies on epipolar error maps, so the final results are heavily influenced by them.
2. The selection of dynamic Gaussians depends on threshold settings, which reduces the generalizability of the pipeline, as determining appropriate thresholds for each scene is not straightforward.

**Questions:**

One of my concerns is that if there is a large viewpoint difference between two frames, then when warping from frame t-1 to frame t, some regions may appear black because certain content in frame t-1 is not visible from the viewpoint of frame t. Would these regions affect the supervision of the TLR loss?

---

> ### Author Response · Authors · 2025-11-25
>
> We sincerely thank the Reviewer J4n4 for the insightful comments. Please find our responses below.
>
> > ### W1. The division between dynamic and static regions relies on epipolar error maps, so the final results are heavily influenced by them.
>
> We acknowledge the importance of separating dynamic and static regions in our method. However, we have found that our approach is not highly sensitive to the threshold $\sigma_1$ used for epipolar error maps within a reasonable range. As demonstrated in Table R1 below, varying the threshold value does not result in significant performance changes. Although there are instances (see Fig. 13(a)) where prior optical flows and epipolar error maps may be inaccurate, our method performs adequately in most scenes. Additionally, we can utilize the Track-Anything [1] foundation model to extract precise dynamic masks, as shown in some previous works [2, 3]. However, this approach relies on user clicks as prompts, which prevents the system from being fully automatic. Therefore, we adopt the strategies of RoDyRF [4] and MoSca [5] by using epipolar error maps derived from optical flows to automatically segment dynamic and static regions without any manual intervention.
>
> **Table R1**: Ablation results on six scenes about the threshold value ${\sigma}_1$ for generating dynamic masks from epipolar error maps. All the metrics listed here represent PSNR.
> | Method | Skateboarder | CheckingEmail | Cleaning | Bridge | | Welding | | Students | |
> |:-------|:------------:|:-------------:|:--------:|:------:|:---:|:-------:|:---:|:--------:|:---:|
> | | LDR-OE | LDR-OE | LDR-OE | LDR-OE | HDR | LDR-OE | HDR | LDR-OE | HDR |
> | $\sigma_1=5e^{-6}$ | 27.91 | **28.08** | 26.19 | 32.74 | **27.56** | 33.63 | 45.99 | **31.53** | 32.02 |
> | $\sigma_1=1e^{-5}$ **(Ours)** | **27.93** | 27.89 | **26.24** | 32.93 | 27.43 | **33.66** | **46.06** | 31.46 | **32.11** |
> | $\sigma_1=5e^{-5}$ | 27.77 | 27.72 | 26.09 | **33.07** | 27.37 | 33.51 | 45.93 | 31.24 | 31.77 |
> | $\sigma_1=1e^{-4}$ | 27.45 | 27.87 | 25.93 | 32.98 | 27.48 | 33.35 | 45.78 | 31.22 | 31.71 |
>
> > ### W2. The selection of dynamic Gaussians depends on threshold settings, which reduces the generalizability of the pipeline, as determining appropriate thresholds for each scene is not straightforward.
>
> Our method involves two important thresholds: ${\sigma}_1$ for generating dynamic masks from epipolar error maps, and ${\sigma}_2$ for dynamic/static Gaussian selection ($N_d/N_f$ in Eq. (1)). To study the influence of these two thresholds, we conduct ablation experiments on six scenes from the Real-Exp-3 and Syn-Exp-3 datasets. The results are presented in Table R1 above and Table R2 below. We can observe that our method is not very sensitive to these two thresholds within a reasonable range. The default values used in our experiments (${\sigma}_1=1e^{-5}$ and ${\sigma}_2=0.1$) perform well across most scenes. Users may adjust these thresholds slightly around the default values if needed for specific scenes; however, our method generally demonstrates robustness to these settings.
>
> **Table R2**: Ablation results on six scenes about the threshold value ${\sigma}_2$ for dynamic/static Gaussian selection ($N_d/N_f$ in Eq. (1)). All the metrics listed here represent PSNR.
>
> | Method | Skateboarder | CheckingEmail | Cleaning | Bridge | | Welding | | Students | |
> |:-------|:------------:|:-------------:|:--------:|:------:|:---:|:-------:|:---:|:--------:|:---:|
> | | LDR-OE | LDR-OE | LDR-OE | LDR-OE | HDR | LDR-OE | HDR | LDR-OE | HDR |
> | $\sigma_2=0.05$ | 27.85 | 27.82 | 26.08 | **32.94** | **27.48** | 33.61 | 45.92 | 31.33 | 31.98 |
> | $\sigma_2=0.1$ **(Ours)** | 27.93 | **27.89** | **26.24** | 32.93 | 27.43 | **33.66** | **46.06** | **31.46** | **32.11** |
> | $\sigma_2=0.2$ | **27.96** | 27.88 | 25.96 | 32.77 | 27.26 | 33.47 | 45.84 | 31.12 | 31.70 |
> | $\sigma_2=0.3$ | 27.62 | 27.75 | 25.95 | 32.59 | 27.33 | 33.39 | 45.71 | 31.03 | 31.68 |

---

> > ### Author Response · Authors · 2025-11-25
> >
> > > ### Q1. One of my concerns is that if there is a large viewpoint difference between two frames, then when warping from frame t-1 to frame t, some regions may appear black because certain content in frame t-1 is not visible from the viewpoint of frame t. Would these regions affect the supervision of the TLR loss?
> >
> > In our implementation, we utilize visibility masks derived from optical flow and depth order to handle occlusions during the computation of TLR loss, as illustrated in Fig. 4. Specifically, when warping from frame $t-1$ to frame $t$, we apply a visibility mask that indicates which pixels in frame $t$ are visible in frame $t-1$. This approach effectively excludes any regions that appear black due to occlusions or significant viewpoint differences from the TLR loss computation. By doing so, we ensure that only valid and visible regions contribute to the temporal luminance regularization, thereby mitigating the potential negative impact of invisible areas on the supervision process. With this TLR loss formulation, the learned dynamic content during well-supervised timeframes can propagate to poorly-supervised timeframes, resulting in a temporally consistent HDR appearance.
> >
> > ### References
> > [1] Jinyu Yang et al., Track Anything: Segment Anything Meets Videos, arXiv:2304.11968, 2023.
> >
> > [2] Qianqian Wang et al., Shape of Motion: 4D Reconstruction from a Single Video, ICCV, 2025.
> >
> > [3] Jongmin Park et al., SplineGS: Robust Motion-Adaptive Spline for Real-Time Dynamic 3D Gaussians from Monocular Video, CVPR, 2025.
> >
> > [4] Yu-Lun Liu et al., Robust Dynamic Radiance Fields, CVPR, 2023.
> >
> > [5] Jiahui Lei et al., MoSca: Dynamic Gaussian Fusion from Casual Videos via 4D Motion Scaffolds, CVPR, 2025.

---

### Official Review · Reviewer_GHAo · 2025-10-31

**Soundness:** 3
**Presentation:** 3
**Contribution:** 3
**Rating:** 6
**Confidence:** 3

**Summary:**

HDR-4DGS presents a well-motivated approach for reconstructing renderable 4D HDR scenes from unposed monocular LDR videos with alternating exposures. The authors propose a two-stage Gaussian Splatting optimization framework, where dynamic HDR video Gaussians are first learned in orthographic camera space and then transformed to world space for joint optimization with camera poses. The method is complemented by temporal luminance regularization, ensuring temporal consistency of HDR appearance. The experimental evaluation is thorough, including both synthetic and real-world datasets, and demonstrates that HDR-4DGS outperforms  state-of-the-art methods.

**Strengths:**

1. This paper presents the first system to address 4D HDR reconstruction from unposed, single-camera, alternating-exposure LDR videos.

2. The proposed two-stage optimization is effective.

3. HDR-4DGS effectively handles varying brightness across frames, which would break conventional photometric reprojection losses.

4. The paper constructs a new benchmark for HDR video reconstruction including real and synthetic scenes.

**Weaknesses:**

1. Although HDR reconstruction is the core contribution, the paper primarily evaluates PSNR/SSIM on tone-mapped images. No HDR-specific metrics (e.g., PQ-PSNR, HDR-VDP).

2. While quantitative results are extensive, the paper provides limited qualitative discussion on typical failure cases.

3. The approach has not been evaluated on low-light, reflective, or transparent surfaces, which may limit applicability in certain real-world conditions.

**Questions:**

I would like to know how the proposed method performs when dealing with scenes that involve extremely fast and complex motion, where motion blur and ambiguity are present.

---

> ### Author Response · Authors · 2025-11-25
>
> We sincerely thank the Reviewer GHAo for the insightful comments. Please find our responses below.
>
> > ### W1.  Although HDR reconstruction is the core contribution, the paper primarily evaluates PSNR/SSIM on tone-mapped images. No HDR-specific metrics (e.g., PQ-PSNR, HDR-VDP).
>
> We have presented the HDR evaluation results, including PSNR, SSIM, and LPIPS, in the $\mu$-law domain, following the methodologies of HDR-NeRF [1], HDR-GS [2], and GaussHDR [3]. Please refer to Table 1 in our paper, which contains a column labeled "HDR" consisting of four sub-columns: PSNR, SSIM, LPIPS, and TAE. It is important to note that we can only quantitatively evaluate the HDR renderings for Syn-Exp-3 scenes with HDR ground truths.
>
> > ### W2.  While quantitative results are extensive, the paper provides limited qualitative discussion on typical failure cases.
>
> We have previously discussed the limitations of our method in Appendix Sec. A.4. In our revised paper, we further expand this section and provide qualitative examples in Fig. 13 that illustrate the typical failure cases mentioned, including: (a) erroneous dynamic/static separation due to inaccurate optical flow estimation; (b) motion blur resulting from rapid camera or object motion; and (c) non-rigid deformation of burning fire. Although these limitations exist, they do not detract from our core contribution of introducing HDR-4DGS for general monocular 4D HDR reconstruction. Our method demonstrates superior performance compared to existing approaches and performs well in most scenarios, as evidenced by extensive quantitative and qualitative results. Furthermore, the identified limitations offer valuable directions for future research.
>
> > ### W3. The approach has not been evaluated on low-light, reflective, or transparent surfaces, which may limit applicability in certain real-world conditions.
> > ### Q1. I would like to know how the proposed method performs when dealing with scenes that involve extremely fast and complex motion, where motion blur and ambiguity are present.
>
> Our experimental datasets encompass challenging scenarios, including scenes with low-light conditions, non-Lambertian (reflective/transparent) surfaces, and fast or complex motions, as illustrated in Fig. 12 of the revised paper. Some of these scenes are also presented in our supplementary videos. Although our method is designed for general HDR reconstruction from varying-exposure LDR frames and does not specifically address these difficult conditions, it still produces reasonable results. To further enhance performance in such challenging settings, we can incorporate specialized techniques from existing literature.
>
> In low-light scenes, such as those depicted in Fig. 12(a)(b), which are often characterized by inherent noise, operating in the RAW domain offers significant advantages. This is evidenced by studies such as RawNeRF [4] and LE3D [5], which are specifically designed for low-light or dark environments. While our current framework processes alternating-exposure videos, we can easily combine our underlying scene representation with RAW-domain color representation to better manage low-light scenarios.
>
> For transparent surfaces in  Fig. 12(c)(d) and reflective surfaces in  Fig. 12(e)(f), we can integrate reflection/refraction modeling as in GaussianShader [6] and TransparentGS [7]. Since our framework adopts Gaussian Splatting as the scene representation, such modeling can be readily incorporated  to improve the handling of these complex surface properties.
>
> In the fast-motion scene depicted in Fig. 12(g), the captured LDR frames often exhibit motion blur. Consequently, regions affected by blur tend to yield similarly blurry reconstructions due to the pixel-level photometric supervision employed. Nevertheless, the overall reconstruction quality remains acceptable. Although deblurring is not the primary focus of our approach, deblurring techniques such as Deblur4DGS [8] could be seamlessly integrated to enable simultaneous deblurring and HDR reconstruction.
>
> We also include a scene, Skateboarder, with complex motion, as shown in Fig. 12(h). This scene contains several types of motion, including a skateboarder moving rapidly nearby, pedestrians walking slowly in the distance, and flowers swaying in the wind. Additional results for this scene are available in our supplementary videos, which demonstrate that our method effectively manages complex motion. This capability can be attributed to our dynamic Gaussian representation and video Gaussian initialization. The former enables each dynamic Gaussian to possess its own motion trajectory, thereby effectively modeling diverse motion patterns within the scene. The latter facilitates the motion learning of dynamic Gaussians by mitigating the interference of camera motion during the video Gaussian stage.

---

> > ### Author Response · Authors · 2025-11-25
> >
> > ### References
> > [1] Xin Huang et al., HDR-NeRF: High Dynamic Range Neural Radiance Fields, CVPR, 2022.
> >
> > [2] Yuanhao Cai et al., HDR-GS: Efficient High Dynamic Range Novel View Synthesis at 1000x Speed via Gaussian Splatting, NeurIPS, 2024.
> >
> > [3] Jinfeng Liu et al., GaussHDR: High Dynamic Range Gaussian Splatting via Learning Unified 3D and 2D Local Tone Mapping, CVPR, 2025.
> >
> > [4] Ben Mildenhall et al., NeRF in the Dark: High Dynamic Range View Synthesis from Noisy Raw Images, CVPR, 2022.
> >
> > [5] Xin Jin et al., Lighting Every Darkness with 3DGS: Fast Training and Real-Time Rendering for HDR View Synthesis, NeurIPS, 2024.
> >
> > [6] Yingwenqi Jiang et al., GaussianShader: 3D Gaussian Splatting with Shading Functions for Reflective Surfaces, CVPR, 2024.
> >
> > [7] Letian Huang et al., TransparentGS: Fast Inverse Rendering of Transparent Objects with Gaussians, SIGGRAPH, 2025.

---

### Official Review · Reviewer_ww4e · 2025-10-31

**Soundness:** 3
**Presentation:** 3
**Contribution:** 3
**Rating:** 8
**Confidence:** 4

**Summary:**

HDR-4DGS tackles 4D HDR reconstruction from unposed monocular LDR videos with alternating exposures. This problem hasn't been tackled exactly before. Their temporal regularization and 2 stage training to model the world shows superior performance qualitatively and quantitatively over adapted baselines.

**Strengths:**

1. The problem setting is well defined and motivated.
2. The paper is well written and clear.
3. The evaluations done are adequate, both quantitatively and qualitatively.
4. The paper comprehensively ablates all the design features showing the importance/visual effect of each modification.

**Weaknesses:**

1. How does the optmization/loss curves look like with that many losses? Would it be possible to show the curves?
2. Are all scenes at 24-30fps? Have the authors tried any more challenging settings like faster motion (ex - moving cars for autonomous driving applications?), non-lambertian surfaces etc? Those would be nice to haves but not necessary of course.

**Questions:**

1. Just curious as to why GaussHDR tends to remove the foreground object altogether. Any insights?

---

> ### Author Response · Authors · 2025-11-25
>
> We sincerely thank the Reviewer ww4e for the insightful comments. Please find our responses below.
>
> > ### W1.  How does the optmization/loss curves look like with that many losses? Would it be possible to show the curves?
>
> We present the loss curves for two scenes, WavingHands and Cars, as illustrated in Fig. 11 of the revised paper. Due to the resetting of Gaussian opacity, certain losses, such as RGB loss, depth loss, and track loss, may exhibit sudden increases at the resetting iterations. For the TLR loss, it is applied only after the completion of Gaussian densification, so it appears during the later training stage. Despite the presence of multiple losses, they function collaboratively and converge effectively. The stability of the optimization process is attributed to several factors. First, we utilize precomputed 2D priors to guide the optimization, providing effective initializations and constraints for geometry and motion. Second, we implement a two-stage optimization strategy with video Gaussian initialization, which enhances the stability and efficiency of world Gaussian optimization.
>
> > ### W2 (1). Are all scenes at 24-30fps?
>
> Yes, the scenes in our datasets are captured at 24-30 fps. Our method also supports high-frame-rate capturing, where the fast motion can be smoothed. In this scenario, we can select key frames to minimize redundancy while preserving motion information.
>
> > ### W2 (2). Have the authors tried any more challenging settings like faster motion (ex - moving cars for autonomous driving applications?), non-lambertian surfaces etc?
>
> Our experimental datasets encompass challenging scenarios, including scenes with fast motion and non-Lambertian (reflective/transparent) surfaces, as illustrated in Fig. 12 of the revised paper. Some of these scenes are also presented in our supplementary videos. Although our method is designed for general HDR reconstruction from varying-exposure LDR frames and does not specifically address these difficult conditions, it still produces reasonable results. To further enhance performance in such challenging settings, we can incorporate specialized techniques from existing literature.
>
> In the fast-motion scene depicted in Fig. 12(g), the captured LDR frames often exhibit motion blur. Consequently, regions affected by blur tend to yield similarly blurry reconstructions due to the pixel-level photometric supervision employed. Nevertheless, the overall reconstruction quality remains acceptable. Although deblurring is not the primary focus of our approach, deblurring techniques such as Deblur4DGS [1] could be seamlessly integrated to enable simultaneous deblurring and HDR reconstruction.
>
> For transparent surfaces in  Fig. 12(c)(d) and reflective surfaces in  Fig. 12(e)(f), we can integrate reflection/refraction modeling as in GaussianShader [2] and TransparentGS [3]. Since our framework adopts Gaussian Splatting as the scene representation, such modeling can be readily incorporated  to improve the handling of these complex surface properties.
>
> > ### Q1. Just curious as to why GaussHDR tends to remove the foreground object altogether. Any insights?
>
> GaussHDR [4] is a static HDR Gaussian Splatting framework. It may (but not definitely) remove foreground objects due to its optimization process, which minimizes the overall photometric reconstruction error. When a dynamic foreground object moves across a wide range within the scene, it appears at different locations in different frames. Consequently, the object may occupy a specific pixel location in one frame, while the corresponding pixel location in other frames displays background content. As a result, the model may "erase" the dynamic object from the scene to achieve a lower overall error. In cases where a dynamic foreground object moves within a limited range, it may still be retained in the reconstruction; however, the model may struggle to accurately represent it, leading to ghosting artifacts. Please refer to Fig. 7 in the revised manuscript, which illustrates several GaussHDR examples with retained (but ghosted) foreground objects.
>
> ### References
> [1] Renlong Wu et al., Deblur4DGS: 4D Gaussian Splatting from Blurry Monocular Video, arXiv:2412.06424, 2024.
>
> [2] Yingwenqi Jiang et al., GaussianShader: 3D Gaussian Splatting with Shading Functions for Reflective Surfaces, CVPR, 2024.
>
> [3] Letian Huang et al., TransparentGS: Fast Inverse Rendering of Transparent Objects with Gaussians, SIGGRAPH, 2025.
>
> [4] Jinfeng Liu et al., GaussHDR: High Dynamic Range Gaussian Splatting via Learning Unified 3D and 2D Local Tone Mapping, CVPR, 2025.

---

> ### Comment · Reviewer_ww4e · 2025-11-26
>
> Thank you for the comprehensive response authors.
> The loss curves make sense and shows meaningful descend; which also reflects visually in the results.
>
> The addition of Fig 13, demonstrating the limitations of this work, definitely strengthens the work further.
>
> All my queries have been answered and I would like to stay at my current rating: 8 - Accept as Poster.

---

### Author Response · Authors · 2025-11-25
**Overview on Paper Changes**

We sincerely thank the reviewers for their appreciation and valuable feedback, which can help us improve the quality of our work.  We have revised the paper accordingly to address their concerns, with key changes summarized below:
* Add loss curves in Appendix Sec. A.6.1 and Fig. 11 for Reviewer ww4e.
* Add some challenging conditions (low-light, non-Lambertian surfaces, fast/complex motion) involved in our datasets in Appendix Sec. A.6.2 and Fig. 12 for Reviewers ww4e and GHAo.
* Expand Appendix Sec. A.4 (Limitations) and add qualitative examples of some failure cases in Fig. 13 for Reviewer GHAo.
* Add ablation tables about the thresholds for dynamic mask generation and dynamic/static Gaussian selection in Appendix Sec. A.6.3 and Tables 6, 7 for Reviewer J4n4.
* Add ablation tables about other exposure schedules in Appendix Sec. A.6.3 and Table 8 for Reviewer vqUK.
* Add ablation tables about the sensitivity to inaccuracies of exposure values in Appendix Sec. A.6.3 and Table 9 for Reviewer vqUK. Add the recovered camera response function curves in Fig. 14.
* Add ablation tables about absent depth/track/flow priors in Appendix Sec. A.6.3 and Table 10 for Reviewer vqUK.
* Add wide-view-change rendering results in Appendix Sec. A.7 and Fig. 15 for Reviewer vqUK. Add corresponding videos in the supplementary document.

Below we provide detailed answers tackling each of the issues raised by the reviewers.

---

### Meta-Review · Area_Chair_n9JU · 2025-12-31

**Summary:**

Reviewers found the paper to make a good contribution with a well-justified and effective framework design, while addressing a new (albeit slightly narrow) problem involving obtaining HDR dynamic 3D Gaussians via alternating exposure videos.

The main reviewer concerns were:
- Potential issues that may arise from a large number of loss function components (ww4e W1, ).
- Concerns about the ability to handle faster motion (ww4e W2, J4n4 Q).
- Lack of evaluation for HDR-specific metrics (GHAo W1).
- Insufficient discussion about failure cases (GHAo W2).
- Insufficient evaluation on real-world conditions, e.g. involving low-light, reflective, or transparent surfaces (GHAo W3).
- Difficulties of separating between static and dynamic regions, which is part of the methodology (J4n4 W1-2).
- Reliance on existing foundation models (vqUK W1 Q3).
- Other non-alternating exposure schedules are not explored (vqUK W2 Q2).
- Concerns about sensitivity to inaccuracies in exposure timings / camera response function (vqUK Q1).

**Reviewer Concerns:**

Most of the reviewer concerns were reasonably resolved. In any case, all the reviewers were already unanimously leaning towards accepting the paper. The AC would have liked to have seen greater novel viewpoint variation in the video results to better assess the quality of the reconstruction, but acknowledges that this was not originally requested by the reviewers.

**Reviewer Scores:**

The original reviewer scores were 8, 8, 6, 6. Reviewers ww4e and vqUK, who had both given 8's, commented that they will retain their ratings. The AC does not expect the scores to change, at least not in a negative direction.

---

### Decision · Program_Chairs · 2026-01-26

Accept (Poster)